nature
ecology & evolution
# Emergence and intensification of dairying in the Caucasus and Eurasian steppes

Ashley Scott [1,2,3], Sabine Reinhold [4], Taylor Hermes [1,2], Alexey A. Kalmykov [5], Andrey Belinskiy[5], Alexandra Buzhilova[6], Natalia Berezina [6], Anatoliy R. Kantorovich[7], Vladimir E. Maslov[8], Farhad Guliyev[9], Bertille Lyonnet[10], Parviz Gasimov [9], Bakhtiyar Jalilov[9], Jeyhun Eminli[9], Emil Iskandarov[9], Emily Hammer[11], Selin E. Nugent [12], Richard Hagan[1,13], Kerttu Majander [1,14], Päivi Onkamo[15,16], Kerkko Nordqvist [17], Natalia Shishlina [18,19], Elena Kaverzneva[18], Arkadiy I. Korolev[20], Aleksandr A. Khokhlov[20], Roman V. Smolyaninov [21], Svetlana V. Sharapova [22], Rüdiger Krause[23], Marina Karapetian[6], Eliza Stolarczyk[23], Johannes Krause [1,2], Svend Hansen [3 ✉], Wolfgang Haak [1,2 ✉] and Christina Warinner [1,2,24 ✉]

Archaeological and archaeogenetic evidence points to the Pontic–Caspian steppe zone between the Caucasus and the Black Sea as the crucible from which the earliest steppe pastoralist societies arose and spread, ultimately influencing populations from Europe to Inner Asia. However, little is known about their economic foundations and the factors that may have contributed to their extensive mobility. Here, we investigate dietary proteins within the dental calculus proteomes of 45 individuals spanning the Neolithic to Greco-Roman periods in the Pontic–Caspian Steppe and neighbouring South Caucasus, Oka–Volga–Don and East Urals regions. We find that sheep dairying accompanies the earliest forms of Eneolithic pastoralism in the North Caucasus. During the fourth millennium BC, Maykop and early Yamnaya populations also focused dairying exclusively on sheep while reserving cattle for traction and other purposes. We observe a breakdown in livestock specialization and an economic diversification of dairy herds coinciding with aridification during the subsequent late Yamnaya and North Caucasus Culture phases, followed by severe climate deterioration during the Catacomb and Lola periods. The need for additional pastures to support these herds may have driven the heightened mobility of the Middle and Late Bronze Age periods. Following a hiatus of more than 500 years, the North Caucasian steppe was repopulated by Early Iron Age societies with a broad mobile dairy economy, including a new focus on horse milking.

During the early to mid-Holocene (ca. 9.0–3.5 thousand years ago (kya)), dairying played a vital role in the development of human food systems across Europe, Africa and Asia[1–8]. Early agropastoral societies raised livestock animals that could provide them with milk, meat, wool, leather and traction[9], and milk rose to prominence as an especially important, nutrient-rich food source. Milk is rich in protein, fat, sugar (lactose), vitamins and minerals, such as calcium[10], and the water content in milk can be relied on in times of drought or scarcity[11,12]. Although milk itself is highly perishable, it can be transformed through microbial fermentation

and other forms of manipulation into more stable products, such as yogurt, butter, ghee, cheese and curds, that can be stored for longer periods in surplus[13–15].

First attested in Anatolia during the seventh and sixth millennia BC[3,6], ruminant dairying subsequently spread to both Europe and Africa by the late sixth millennium BC[4,16], but less is known about its initial dispersals into Asia[17–19]. One major vector by which dairying spread was the Eurasian steppe, an enormous expanse of grasslands stretching 6,000 km from the Carpathian Basin to Mongolia. Recent studies have traced the introduction of dairying in Mongolia

[1]Department of Archaeogenetics, Max Planck Institute for Evolutionary Anthropology, Leipzig, Germany. [2]Department of Archaeogenetics, Max Planck Institute for the Science of Human History, Jena, Germany. [3]Institute for Pre- and Protohistoric Archaeology and Archaeology of the Roman Provinces, Ludwig Maximilian University Munich, Munich, Germany. [4]Eurasia Department, German Archaeological Institute, Berlin, Germany. [5]'Nasledie' Cultural Heritage Unit, Stavropol, Russia. [6]Research Institute and Museum of Anthropology, Lomonosov Moscow State University, Moscow, Russia. [7]Department of Archaeology, Faculty of History, Lomonosov Moscow State University, Moscow, Russia. [8]Institute of Archaeology RAS, Moscow, Russian Federation. [9]Department of Humanitarian and Social Sciences, Institute of Archaeology, Ethnography and Anthropology, Azerbaijan National Academy of Sciences, Baku, Azerbaijan. [10]PROCLAC/UMR 7192 Laboratory, French National Centre for Scientific Research, Paris, France. [11]Near Eastern Languages and Civilizations and Price Lab for the Digital Humanities, University of Pennsylvania, Philadelphia, PA, USA. [12]Faculty of Technology, Design & Environment, Oxford Brookes University, Oxford, UK. [13]Department of Archaeology, University of York, York, UK. [14]Institute of Evolutionary Medicine, University of Zürich, Zürich, Switzerland. [15]Department of Biology, University of Turku, Turku, Finland. [16]Department of Biosciences, University of Helsinki, Helsinki, Finland. [17]Department of Cultures, University of Helsinki, Helsinki, Finland. [18]State Historical Museum, Moscow, Russia. [19]Peter the Great Museum of Anthropology and Ethnography (the Kunstkamera), Saint Petersburg, Russia. [20]Department of History and Archaeology, Samara State University of Social Sciences and Education, Samara, Russia. [21]Lipetsk State Pedagogical University, Lipetsk, Russia. [22]Institute of History and Archaeology, Ural Branch of the Russian Academy of Science, Ekaterinburg, Russia. [23]Department of Archaeological Sciences, Johann Wolfgang Goethe University, Frankfurt am Main, Germany. [24]Department of Anthropology, Harvard University, Cambridge, MA, USA. ✉e-mail: svend.hansen@dainst.de; wolfgang_haak@eva.mpg.de; warinner@fas.harvard.edu

to ca. 3000 BC with the appearance of mobile steppe herders associated with the Early Bronze Age Afanasievo culture[2], a group with close genetic and cultural ties to pastoralists on the Pontic–Caspian steppe, most notably the Yamnaya culture (ca. 3300–2500 BC)[20–23]. Populations from the Pontic–Caspian steppe are also linked to Late Neolithic and Bronze Age westward expansions, including the emergence of the Corded Ware (2900–2200 BC) and Bell Beaker (2750–1800 BC) phenomena in Europe[24–27]. Understanding the population and economic history of the Pontic–Caspian steppe, the source region for these continental-scale expansions during the third millennium BC, is critical for revealing the main factors that drove the heightened mobility of Eneolithic and Early Bronze Age pastoralists in Eurasia.

When Pontic–Caspian steppe populations first began dairying and how their animal management strategies may have influenced their mobility and subsequent migrations remain poorly known. From the Mesolithic through the Eneolithic, populations living in the southern Russian plain and Caucasus region primarily hunted local wild game, which included aurochs (*Bos primigenius*), saiga antelope (*Saiga tatarica*), red deer (*Cervus elaphus*), tarpan (*Equus ferus*), onager (*Equus hemionus*) and wild boar (*Sus scrofa*), as well as birds, fish and molluscs[28–31]. Animal husbandry of domesticated sheep (*Ovis aries*), goats (*Capra hircus*), cattle (*Bos taurus*) and pigs (*Sus scrofa*) spread to the North Caucasian steppe from Anatolia during the fifth millennium BC by either a circum-Pontic route[28] or by crossing the Caucasus mountains from the south[32–35]. By the mid-fifth millennium BC, agropastoralists of the Cucuteni–Trypillia culture in Ukraine were regularly interacting with steppe populations north of the Black Sea[36], and Eneolithic populations genetically related to South Caucasian and Anatolian agropastoralist groups had become established in the North Caucasus piedmont steppe[32,33,37] and were part of a broader Mesopotamian interaction sphere[38,39].

After the introduction of animal husbandry to the region, Bronze Age steppe populations innovated a new economic system of mobile pastoralism focused on sheep and cattle[40], and settlements became effectively absent on the steppe for the next two millennia[40,41]. This new, more mobile form of pastoralism is first evident among Steppe Late Maykop groups (3500–2900 BC), who fall broadly within the Late Maykop cultural sphere but are genetically distinct from their higher-elevation counterparts[33], and fully mobile pastoralism subsequently became the predominant subsistence strategy on the steppe with the Yamnaya culture (3,300–2,500 BC)[41]. Horse domestication occurred during the third millennium on the Pontic–Caspian steppe[42,43], and, by the late third and early second millennium BC, domestic horses were increasingly part of the steppe mobile pastoralist economy[44] and had even spread to Anatolia and Mesopotamia through Pontic–Caspian–Transcaucasian interaction networks[45]. Mobile pastoralism continued among the Catacomb (2800–2200 BC) and North Caucasus Culture (NCC; 2800–2400 BC) groups in the steppe until worsening climatic conditions and aridification ca. 2300–2200 BC, in association with the 4.2 kya climate event[46,47], ultimately led to an abandonment of the steppe region by 1700 BC[40,41]. Despite their cultural differences, recent palaeogenomic analysis has shown that these Bronze Age steppe populations were genetically highly similar[33], which may, in part, reflect their mobile lifestyles and persistent multicultural interactions over millennia[40].

Throughout the Pontic–Caspian steppe, sheep, goat and cattle dominate most studied steppe archaeofaunal collections from the fourth to second millennia BC[41,48,49]. Wheeled transport in the form of wagons first appears in kurgans (burial mounds) of the Steppe Late Maykop in the second half of the fourth millennium BC[50], and such technology is argued to be essential for enabling the household mobility required for mobile pastoralism[40]. Oxen teams dated to the same period and, later, horses and chariots in the second millennium BC, further facilitated mobility[51]. Sheep wool was present in the North Caucasus by the early third millennium BC, possibly having originated in Anatolia, and the use of wool subsequently spread across the steppe and into Inner Asia during the second millennium BC[52]. Among the region's major secondary products, dairying is argued to have possibly emerged first[50], in part because dairying was already well established in both Anatolia and surrounding regions by the sixth millennium BC[6,53–55], whereas evidence for traction and wool are only attested millennia later. Nevertheless, current evidence for early dairying in the Pontic–Caspian steppe is, until now, only attested on its eastern fringes[7]. Previous isotopic studies have been unable to identify clear indications of dairy consumption, finding instead non-specific evidence for high consumption of animal protein and a highly complex isoscape, reflecting both ecological diversity and temporal climatic shifts[41,48,56]. However, the isotopic data suggest a stronger contribution of sheep or goat products to the human diet than those from cattle[41]. Few zooarchaeological studies have systematically investigated herd management and mortality profiles in the region, but the earliest agropastoralist communities in the North Caucasus piedmont steppe were not thought to have engaged in dairying[49]. Likewise, there are few indications of animal management for milk production among Neolithic agropastoralist communities in the South Caucasus[42]. Rather, it is only in the second millennium BC that zooarchaeological studies from Late Bronze Age settlements in the Caucasus have found clear evidence for the deliberate keeping of sheep for milk production[57,58], and it is only later during the Iron Age that cattle show mortality profiles consistent with dairying[59].

The absence of settlements on the steppe and the near-exclusive archaeological focus on mortuary contexts have made it difficult to reconstruct the nature and extent of dairying in the wider North Caucasian pastoralist economy. In this article, we apply high-resolution tandem mass spectrometry to human dental calculus from 45 individuals at 29 sites in the North Caucasus ($n = 27$) and the neighbouring South Caucasus ($n = 9$), Oka–Volga–Don ($n = 7$) and East Urals ($n = 2$) regions (Fig. 1a,b, Supplementary Data 1 and Supplementary Information) to identify evidence of consumed dairy proteins in populations spanning the Neolithic to the Greco-Roman periods (ca. 6000 BC to 200 AD). We find that dairy products were consumed in the North Caucasus from the late fifth millennium BC onwards and that a dairy-inclusive subsistence characterizes even the Eneolithic populations in the piedmont and steppe zones. Dairy consumption was prevalent for all analysed periods and ecotones in the North Caucasus, with milk proteins identified in 26 of 27 tested individuals. We identify an initial, near-exclusive dairying focus on sheep among the Maykop, Steppe Maykop and early Yamnaya, followed by diversification within the late Yamnaya, NCC and Catacomb cultures during the Middle Bronze Age to additionally incorporate goat and cattle milking.

**Fig. 1 | Map and timeline of sites and individuals in the study and milk protein results. a**, Map of study area and major cultural regions mentioned in the text: Oka–Volga–Don, East Urals, North Caucasus, South Caucasus and Anatolia. Extent of the Pontic–Caspian steppe is shown in grey. Inset: enhanced view of North Caucasus sites. **b**, Timeline of sites and individuals analysed in this study. Individuals are organized by region, with archaeological culture or period indicated by colour corresponding to the legend. White circles indicate median calibrated radiocarbon dates, and error bars are 2 s.d. Coloured bars display the time spans conventionally associated with the archaeological cultures and time periods. **c**, Milk protein evidence by individual, displayed as total PSM count to the milk proteins BLG, alpha-lactalbumin and alpha-S1-casein. Consensus livestock assignment was determined by parsimony.
[a]Two dental calculus samples were analysed from ZO2002. Basemap is from https://www.naturalearthdata.com/.

Later, during the Early Iron Age, we observe direct evidence of horse milk consumption in association with pre-Scythian groups repopulating the steppe after a centuries-long hiatus. In the South Caucasus, we identify evidence of cattle milking (ca. 3700 BC) nearly 1,000 years before we first observe it in the North Caucasus (ca. 2700 BC), and, in the Oka–Volga–Don region, we observe

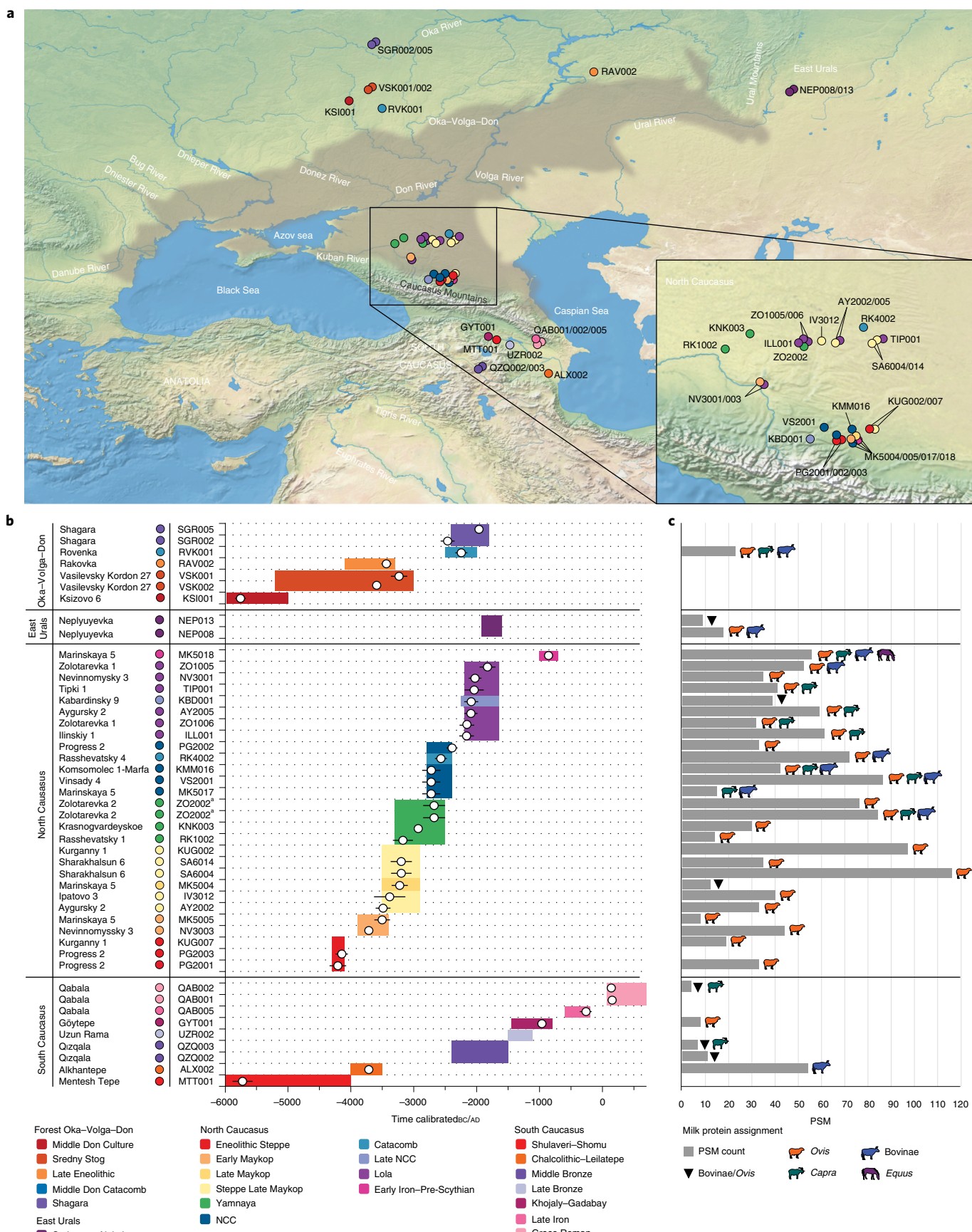

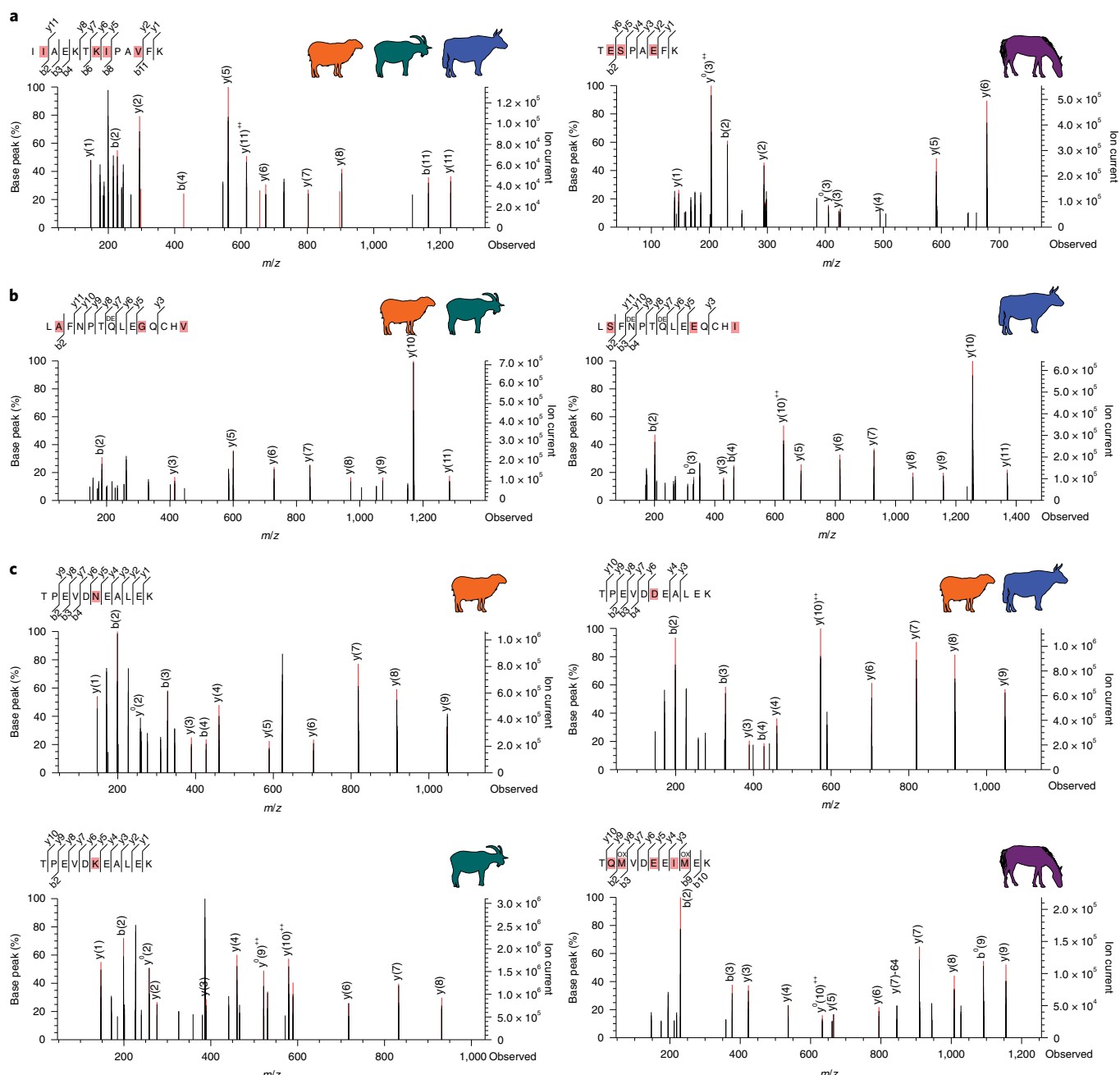

**Fig. 2 | Representative tandem mass spectrometry spectra of selected BLG peptides with differing levels of taxonomic resolution observed in this study.**
**a**, Overall, most BLG sequences were highly conserved among bovids (left) but distinct from equids (right). Spectra originate from AY2005 and MK5018.
**b**, Among bovids, the BLG C-terminus peptide distinguishes caprines (left) and bovines (right). Spectra originate from VS2001 and VS2001. **c**, The most frequently observed peptide reliably distinguishes *Ovis* (upper left), *Capra* (lower left) and *Equus* (lower right) but cannot distinguish *Ovis* and Bovinae due to the ambiguity of the sixth residue, which may be aspartic acid (Bovinae) or deamidated asparagine (*Ovis*)[6] (upper right). Spectra originate from KUG007, RK4002, VS2001 and MK5018. The b- and y-ion series is shown at the top left of each spectrum, and taxonomically informative residues within the peptide sequence are highlighted in pink. A comprehensive list of all identified PSMs and taxonomic assignments is provided in Supplementary Data 3.

limited evidence of dairying, beginning only during the second millennium BC.

## Results
Milk proteins were identified in 34 of 45 analysed individuals across all time periods (Fig. 1c and Supplementary Data 1). Protein recovery in 31 individuals was sufficient to allow the identification of major ruminant livestock milks from sheep (*Ovis*), goat (*Capra*) and/or cattle (*Bos*/Bovinae), whereas the milk proteins

of three individuals were represented by non-specific bovid peptides, indicating either sheep or cattle. Additionally, one individual had taxonomically distinctive peptide spectral matches (PSMs) to *Equus* milk proteins. Beta-lactoglobulin (BLG), which was detected for all dairy livestock (Fig. 2), was the most prevalent and abundant milk protein detected, a pattern consistent with previous studies of dental calculus[2,6,60]. In addition to BLG, which was identified in all 34 milk-positive individuals, we also identified the whey protein alpha-lactalbumin in two individuals and the

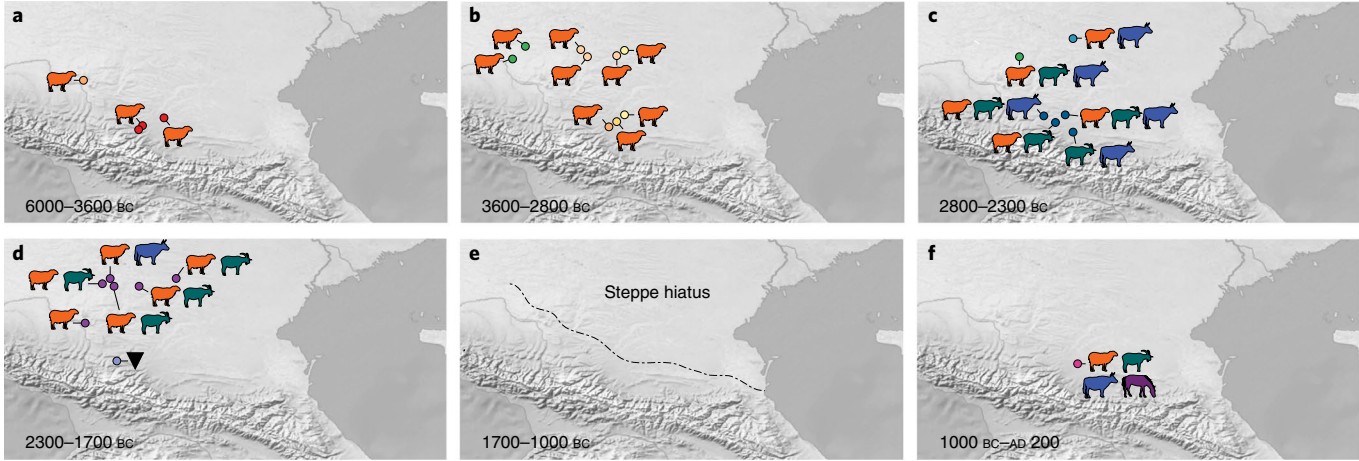

**Fig. 3 | Changing dairy patterns through time in the North Caucasus region. a**, During the Eneolithic and initial Bronze Age, dairying focused on sheep in the North Caucasus from 4200 BC onwards. **b**, Sheep dairying continued during the Early Bronze Age among the Maykop, Steppe Maykop and early Yamnaya. **c**, After 2800 BC, goat and cattle dairying appeared for the first time in the steppe, and diversified dairy economies of sheep, goats and cattle characterize the late Yamnaya, NCC and Catacomb cultures. **d**, Diversified dairy economies persisted among the post-Catacomb and Lola cultures, but with an increased focus on sheep and goats as environmental conditions declined. **e**, During the Late Bronze Age, the North Caucasus steppe was largely depopulated, and ca. 1700 BC a centuries-long hiatus began that corresponded to a period of extreme aridity. Dashed line shows the southern extent of depopulation. **f**, After 1000 BC, post-hiatus groups repopulated the steppe in the Early Iron Age, resuming sheep, goat and cattle milking and also introducing horse milking. Site colours and animal symbols correspond to those in Fig. 1. All tested individuals in the map extent are shown, including those without evidence of milk protein.

curd protein alpha-S1-casein in two individuals. All dental calculus samples yielded proteomes consistent with an oral microbiome profile, and age-associated N/Q deamidation was a top modification across the dataset (Supplementary Data 2). Milk protein peptide-level deamidation is reported in Supplementary Data 3. No dietary proteins were detected in non-template extraction controls (Supplementary Data 3).

**North Caucasus.** North of the Caucasus mountains, within a geographically and culturally contiguous region that encompasses the piedmont zone, steppe and southern Russian plain, we analysed dietary proteins within the dental calculus proteomes of 27 individuals, including three Eneolithic, 23 Bronze Age and one Early Iron Age individual. Overall, we identified milk proteins in 96% of individuals ($n = 26$) (Fig. 1c) and observed high levels of milk protein PSMs per individual (mean $47 \pm 27$; Supplementary Data 3), with milk peptides often being among the most abundant peptides identified in the dental calculus proteomes. Among Eneolithic individuals, two of three were positive for milk proteins. The oldest individual from this region in our study, PG2001 from the piedmont site of Progress 2 and dated to 4338–4074 BC, indicates that dairying has been a feature of the region's economy since at least the late fifth millennium BC. During the fourth and third millennia BC, we observed a continued reliance on dairying among all analysed Maykop and Steppe Maykop individuals (ca. 3900–2900 BC; $n = 7$), both in the piedmont and steppe zones as well as in all Yamnaya individuals (ca. 3300–2500 BC; $n = 3$). Notably, we detected only *Ovis* milk proteins at Eneolithic, Early Maykop, Late Maykop, Steppe Late Maykop and early Yamnaya sites, suggesting that dairying was a specialized activity focused on sheep during the fourth and fifth millennia BC (Fig. 3a,b). At the start of the early third millennium BC, we identified a broad shift in pastoralist practices towards more diversified dairying based on sheep, goat and cattle milk (Fig. 3c,d). Milk proteins from these three ruminant species were identified among individuals associated with the late Yamnaya (ca. 2850–2500 BC; $n = 1$), NCC (ca. 2800–2400 BC; $n = 4$), Catacomb (ca. 2800–2400 BC; $n = 1$), late NCC (ca. 2200–1650 BC, $n = 1$) and Lola/post-Catacomb (ca.

2200–1650 BC; $n = 6$) cultures, with most individuals having consumed the dairy products of two or three different animal milks in the form of sheep and goat milk, sheep and cattle milk or sheep, goat and cattle milk (Fig. 1 and Supplementary Data 3). Finally, during the Early Iron Age (eighth–fifth centuries BC), we observed the incorporation of horse (*Equus*) milk into the dairy economy (Fig. 3e), with *Ovis*, *Capra*, *Bos* and *Equus* milk proteins identified in the dental calculus of individual MK5018.

**South Caucasus.** In the South Caucasus, we analysed dietary proteins within the dental calculus proteomes of nine individuals dating from the Neolithic to Greco-Roman periods and identified milk proteins in half of the analysed individuals (Fig. 1c and Supplementary Data 3). No milk proteins were detected in the earliest individual, MTT001, dated to 5879–5562 BC from the Neolithic site of Mentesh Tepe associated with the Shomutepe–Shulaveri Culture. However, milk proteins were detected from the fourth millennium BC onwards in individuals dating to the Chalcolithic at Alkhantepe ($n = 1$), the Middle Bronze Age at Qızqala ($n = 2$), the Iron Age at Göytepe ($n = 1$) and the Greco-Roman period at Qabala ($n = 1$). Unlike in the North Caucasus, we did not observe an early focus on sheep dairying; rather, the earliest detected milk protein, identified in individual ALX002 dating to 3776–3651 BC, was assigned to cattle (Bovinae). Overall, we identified cattle (Bovinae), goat (*Capra*) and sheep (*Ovis*) milk protein in the South Caucasus but no horse (*Equus*) milk in any period there (Fig. 3).

**Oka–Volga–Don region.** In the Oka–Volga–Don region, we analysed dietary proteins within the dental calculus proteomes of seven individuals dating from the Eneolithic through the Middle Bronze Age (Fig. 1c and Supplementary Data 3). Despite excellent protein recovery, no milk proteins were detected in an individual from the Neolithic–Bronze Age site of Ksizovo 6, dating to 5837–5670 BC, nor from individuals associated with the Sredny Stog culture ($n = 2$) at the Eneolithic–Bronze Age site of Vasilevsky Kordon 27, dating to ca. 3600–3100 BC. Milk proteins were also absent from individual RAV002, dating to 3514–3356 BC, and from two Middle Bronze

Age individuals from the Shagara cemetery, dating to 2572–1893 BC. Only an individual at the site of Rovenka tested positive for milk proteins. This individual, RVK001, was associated with a late Catacomb culture site, dating to 2339–2148 BC, and was positive for sheep (*Ovis*), goat (*Capra*) and cattle (Bovinae) milk proteins (Fig. 1c).

**East Urals region.** We analysed two individuals from the East Urals region at the site of Neplyuyevka associated with the Late Bronze Age Srubnaya–Alakul cultural variant and dating to ca. 1900–1600 BC (Fig. 1c). We detected milk proteins for both individuals, identifying sheep (*Ovis*) and cattle (Bovinae) peptide sequences for NEP008 and non-specific bovid peptide sequences indicating either sheep or cattle for NEP013 (Supplementary Data 3).

## Discussion
**Eneolithic populations practiced dairy pastoralism.** Our results provide robust evidence that sheep dairying was practiced among fifth millennium BC Eneolithic groups in the North Caucasus piedmont and steppe zones. This finding resolves long-standing questions about the antiquity of dairying in the North Caucasus[47], as well as the species focus of early dairy herds, and it contributes to a growing body of evidence that dairying was likely introduced with domesticated livestock into the North Caucasus from the south. Recent palaeogenomic studies identified a genetic cline connecting Neolithic populations in eastern Anatolia and the South Caucasus that likely formed as early as 6500 BC[32], and continued population interaction into the Chalcolithic and Early Bronze Age periods (5500–3000 BC) suggests that these regions maintained close contact, with animal husbandry focused on pigs and ruminants also spreading via this corridor[61,62]. Early agropastoralists living in the northern Caucasus foothills associated with the Darkveti–Meshoko Eneolithic culture (ca. 4500–4000 BC) have a clear genetic connection to populations south of the Caucasus exhibiting Anatolian ancestry[33], suggesting a trans-Caucasian population expansion.

Although it has been speculated that dairying may have spread to the North Caucasus via these southern connections[50], few systematic zooarchaeological studies have been conducted, and the Eneolithic/Chalcolithic layers at the piedmont site of Meshoko Cave, which are among the best studied for the period[49], have yielded limited faunal remains, primarily of pigs, sheep, goats and cattle slaughtered at various ages. Subsequent attempts to clarify the agropastoralist economy using stable isotope analysis[41,48] have yielded equivocal results as to whether dairying was an Eneolithic or Bronze Age innovation in the North Caucasus. Here, through the identification of taxonomically informative peptides from the milk-specific protein BLG, we confirmed sheep milk consumption by Eneolithic individuals at the sites of Progress 2 and Kurganny 1. Notably, we found that dairy consumption was evident among individuals lacking Anatolian ancestry, such as PG2001[33], demonstrating that the adoption of dairying by North Caucasian transitional foragers was already underway during the late fifth millennium BC, which precedes Yamnaya expansions by a millennium.

**Maykop and Steppe Maykop dairy focused on sheep not cattle.** With the start of the fourth millennium BC, we found a continued reliance on dairy pastoralism revealed by ubiquitous evidence of milk consumption among all tested Maykop and Steppe Maykop individuals, further clarifying the high dependence of these groups on animal products[41,47]. Surprisingly, however, the dairy economy retained an apparent focus on sheep. Although sheep are known to have been important livestock for these groups[40,47], cattle feature more prominently at Maykop mortuary sites. They are modelled into gold and silver figurines[63], and an emphasis on the power and mobility of cattle is visible in funerary offerings of cattle crania, yokes and nose rings representing oxen teams[50]. Cattle also appear

in bone assemblages at Maykop settlements[49]. The perishability of the major sheep secondary products of milk and wool, in contrast to the high visibility of cattle-associated material culture and skeletal remains, may have contributed to a biased understanding of the relative importance and roles of these livestock at Early Bronze Age sites[64]. Our results suggest that cattle were not important dairy livestock during this period and that there was probably a sharp division in livestock use among the Maykop and Steppe Maykop groups[41], with sheep being the primary targets of dairying and cattle mainly being used for traction and as a signifier of social identity and status.

**Dairy livestock diversified during Middle Bronze Age.** A change in dairying strategy to focus on more livestock species coincides with the Yamnaya horizon. Following the Maykop period, mobility expanded ever further with Yamnaya groups, who became the first permanently mobile pastoralists[17,44,65,66]. Although two early Yamnaya individuals analysed here yielded evidence of only sheep milk product consumption, a more diversified profile comprising sheep, goat and cattle milk was observed for a late Yamnaya individual at the site of Zolotarevka (ZO2002). This trend towards reliance on a broader range of dairy livestock continued and intensified during the Middle Bronze Age, when we observed a general diversification of pastoralist diets to include sheep, goat and cattle milk routinely. Most individuals of the Middle Bronze Age Catacomb, NCC, Late NCC and Lola cultures tested in this study consumed the dairy products of two or three livestock species. Palaeoecological studies have indicated that climate began to shift during the late Yamnaya phase, which also coincided with the first appearance of the Catacomb and NCC groups[48]. Before this, the climate experienced by the Maykop, Steppe Maykop and early Yamnaya was more favourable[67,68] and conducive to regular, short-distance annual mobility[47,48]. Subsequent aridification encouraged increased mobility, resulting in the exploitation of a wider range of steppe environments beyond the traditional Yamnaya cultural sphere to support livestock herds[40,48]. The shift to more diverse dairy herds in the North Caucasus also overlaps in time with Yamnaya expansions into southeastern Europe, as well as the parallel rise and expansion of the Corded Ware complex across northeastern and central Europe[27], suggesting that these events may be related to broader changes occurring within steppe and forest–steppe pastoralist societies at the time. Our results suggest that an initial diversification of production strategies to include sheep, goat and cattle milk may have functioned as an adaptation to an increasingly turbulent ecological setting, but this subsequently led to overgrazing and lasting damage to pastures due to ground compaction, soil nutrient loss and decreasing plant biomass[48,69]. At the end of the third millennium BC, coinciding with the emergence of the Lola culture, an intensified drought caused deflation and salinization of the soils in the already dwindling regional watersheds[40,69]. During the Lola period, water-demanding cattle may have decreased in dairying importance from the preceding Catacomb and NCC periods, as only one of six Lola individuals yielded evidence for cattle milk consumption. After 1700 BC, the steppe and piedmont zones of the Northern Caucasus appear to have been largely depopulated until the ninth or eighth century BC[57,70,71], whereas pastoralist groups continued to occupy the high plateaus of the Caucasus Mountains[72].

**Post-Bronze Age adoption of horse milking.** In our dataset, we found no evidence of horse milk consumption until the ninth century BC, when Early Iron Age groups repopulated the North Caucasus steppe and piedmont zones[33,41]. Horses are well adapted to steppe environments, and recent palaeogenomic research has identified the lower Don–Volga region, possibly as early as the mid-sixth millennium BC, as the domestication centre of the DOM2 horses that characterize present-day lineages[43,45]. From the Pleistocene until

the Bronze Age, horses were hunted on the Pontic–Caspian steppe and have long been symbolically represented in figurines and ritual deposits[28,73]. Horses are also useful for steppe pastoralists because of their digging (*tebenevka*) reflex, which allows them to graze through thick snow deposits, thereby opening up winter pasture for ruminants[48,74,75]. In the North Caucasus, skeletal remains of the ancestors of DOM2 horses are sporadically found in steppe kurgans from the Late Maykop period onwards[43], but the role of horses in these pastoralist societies is unclear. The first undisputed evidence of horse traction dates to ca. 2000 BC at the site of Sintashta east of the Urals, where elaborate horse chariot burials have been found in Middle and Late Bronze Age kurgans[51,76,77]. Earlier Bronze Age wagons, such as those associated with the Late Maykop, Yamnaya and Catacomb cultures, had been pulled by oxen teams[50]. Herding on horseback, which may have begun ca. 2200 BC with the selection of traits suitable for riding[43], would have enabled individual pastoralists to control more livestock at one time and to access pastures across a wider area[75]. Later, horses became particularly prominent in the archaeological record of Early Iron Age Scythians and Sarmatians, who used horses for cavalry[78,79]. In addition to traction and riding, horses can also be exploited for milk, which is traditionally fermented to produce an alcoholic beverage in contemporary Eurasian steppe cultures[80,81]. However, the origin of horse milking is not known. Isotopic evidence from lipids in pottery suggests that Przewalski's horses, reflecting a separate domestication lineage (DOM1)[76], may have been milked as early as the mid-fourth millennium BC at the site of Botai in northern Kazakhstan[76,82]. It is unclear what, if any, influence early milking at Botai had on the management of DOM2 lineages, the ancestors of modern domestic horses. Currently, the earliest proteomic evidence of horse milk consumption comes from two individuals with problematic dates at the Bronze Age site of Kriviyansky IX in the Lower Don region[7] and, later, at the Late Bronze Age site of Uliastai Dood Denzh located in Mongolia, where the dental calculus of an individual dated to ca. 1200 BC with Sintashta-related ancestry yielded evidence of horse milk proteins[2,20]. Despite an apparent early presence of horse milking at Kriviyansky IX, dating to the third, or possibly fourth, millennium BC, we found no other evidence of horse milking in the North Caucasus region during the Early, Middle or Late Bronze Age. Rather, its late appearance in our dataset suggests that horse milking was a highly limited activity while diverse domestication pathways unfolded, and horses were used for various purposes. Horse milking may have been permanently established in the northern Caucasus only after a later reintroduction by pre-Scythian groups during the first millennium BC. Greek texts, such as *The Iliad*, later referred to these pre-Scythian steppe nomads as horse milk drinkers[83].

**Macroregional perspectives on the spread of dairying.** The Pontic–Caspian steppe has long been recognized as a major centre for pastoralist innovation. Here we show that dairying was an early and enduring feature of the pastoralist economy not only in the Northern Caucasus, but also in the South Caucasus. In our dataset, we observed the earliest evidence of milk consumption in the South Caucasus at Alkhantepe, a Late Chalcolithic site with Leilatepe ceramics[84,85]. The contemporaneous Leilatepe and Early Maykop cultures share many features[39,86], but we found that the agropastoralists at Alkhantepe were milking cattle, whereas we observed only sheep milking at Early and Late Maykop sites in the north. Sheep and cattle have different ecological needs, and, in particular, sheep require less water and can survive harsher winters than cattle. As such, environmental factors may have played a role in influencing the selection of dairy livestock in these two regions. During the third millennium BC, it is known that the economic importance of pastoralism increased in the South Caucasus, especially during the Kura–Araxes period[56,87], but we did not have corresponding samples to examine this. Although steppe cultural elements, such as kurgans

(burial mounds), had been present in the South Caucasus since the Late Chalcolithic[88], kurgans greatly increased during the Middle Bronze Age[89], and we next observed dairy product consumption at the Middle Bronze Age fortified agropastoral site of Qızqala, with ruminant dairy proteins present in both individuals analysed for this study. Although Middle Bronze Age cultures in both the North and South Caucasus largely became fully mobile to support their herds[90], the inhabitants of Qızqala relied on a more flexible subsistence strategy that included both settlement occupation and seasonal movement of livestock[89,91]. Our results show a reliance on dairy technology for subsistence for these mobile pastoralists. Next, we found evidence of sheep milk consumption by one individual from an intrusive Late Bronze/Early Iron Age burial associated with the Khojaly–Gadabay culture at the Neolithic site of Göytepe. This is the earliest unequivocal evidence of sheep milking in our South Caucasus dataset. Later, during the Greco-Roman era, we observe evidence of sheep, goat and cattle milk at Qabala, a site associated with complex and intensive agriculture as well as with local herding.

Despite cultural interaction with adjacent communities of the Pontic–Caspian steppe, communities in the Oka–Volga–Don forested regions maintained economies based on hunting, gathering and fishing that were particularly suited to local ecozones. Stable isotope studies suggest that this was the prevailing economic strategy until the end of the third millennium BC during the Middle Bronze Age[48,92,93] when Oka–Volga–Don communities transitioned to agropastoralist subsistence[44]. Although populations further to the east, between the Volga River and the Ural Mountains, practiced ruminant dairying from ca. 3000 BC onwards[7], the near-complete lack of evidence for ruminant milk consumption from the seven individuals representing the Oka–Volga–Don region in our study is consistent with a late introduction of ruminant dairying west of the Volga, despite the fact that domesticated animals were introduced in small quantities during the late fourth millennium BC. Here, only one Catacomb-associated individual with cultural links to the steppe zone, recovered from the site of Rovenka, yielded ruminant milk proteins, which were sourced from sheep, goats and cattle.

In parallel to the expansion of pastoralism to the forest–steppe zone, contact and admixture with late farming groups in eastern Europe, such as Cucuteni–Trypillia and Globular Amphora, resulted in a mixed form of agropastoralism with heavy reliance on pastoralism[94], followed by a subsequent eastward expansion of the Corded Ware complex during the third and early second millennia BC, which is also attested by archaeogenetic data[22,95]. This sphere of influence includes Fatyanovo/Balanovo and subsequent Abashevo, Sintashta, Andronovo, and Srubnaya groups[94], and individuals associated with these cultures share very similar genetic profiles. We analysed two individuals linked to the Srubnaya culture at the Middle to Late Bronze Age site of Neplyuyevka in the region east of the Ural Mountains and identified evidence of ruminant milk consumption. Future work combining palaeogenomic and palaeodietary research could help to better clarify the relationships between these populations and the nature and spatio-temporal patterning of dairy technologies in this region.

## Conclusion
Proteomic analysis of human dental calculus has revealed a dynamic trajectory of dairy pastoralism in the North Caucasus steppe and adjacent regions from the Eneolithic to the Greco-Roman periods. Dairying was integral for the spread of animal husbandry by groups crossing the Caucasus mountains from south to north during the Eneolithic, and it was quickly adopted and further developed into an effective and sustainable technology—dairy pastoralism—by neighbouring steppe communities. This innovation forms the basis of the Eurasian steppe lifestyle that continues until today. Initial pastoralist strategies focused on sheep dairying and cattle traction, whereas fully mobile pastoralism arose for the first time during the Yamnaya

period. Deteriorating climatic conditions challenged steppe herders during the Middle and Late Bronze Ages, who responded by diversifying their set of dairying livestock and expanding their herding range, until the steppe was ultimately abandoned in the mid-second millennium BC. Later, following a centuries-long hiatus, the steppe was repopulated by Early Iron Age pastoralists who practised horse milking. The turbulent third millennium BC, during which vast stretches of Eurasia experienced social and demographic upheaval, is now coming into sharper focus. Climatic pressures and the needs of dairy herds altered how pastoralists used the North Caucasus steppe and may have contributed to the heightened mobility of third-millennium-BC steppe herders, whose descendants spread across Eurasia within the span of only a few centuries. Future research on the genomes of ancient dairying livestock and additional dental calculus proteomes from adjacent steppe populations north of the Black Sea and east of the Urals will help to further clarify the origins and dispersals of dairying breeds and practices that promoted the lasting cultural and subsistence traditions that reshaped the Eurasian steppe zone and profoundly transformed the Bronze Age Eurasian world.

## Methods

**Sampling.** Dental calculus sampling was performed on site at archaeological institutions and museums and in a dedicated ancient biomolecules laboratory at the Max Planck Institute for the Science of Human History (MPI-SHH). Disposable nitrile gloves were worn during collection, and calculus was sampled using dental curettes that were replaced or cleaned with isopropanol between samples. Calculus was collected onto weighing paper and stored in microcentrifuge tubes. Samples were further analysed at the MPI-SHH ancient proteomics laboratory, where they were weighed and subsampled before protein extraction. Approximately 5–13 mg of dental calculus was used for each protein analysis.

**Radiocarbon dating.** A total of 24 new radiocarbon dates were obtained by accelerator mass spectrometry of bone and tooth material at: the Curt-Engelhorn-Zentrum Archäometrie in Mannheim, Germany; the Finnish Museum of Natural History (Hela) in Helsinki, Finland; the Oxford Radiocarbon Accelerator Unit in Oxford, United Kingdom; and the Russian Academy of Sciences in Moscow, Russia. Uncalibrated dates were successfully obtained for all but one tested sample (Supplementary Data 1). An additional 21 previously published radiocarbon dates for individuals in this study were also compiled and analysed, making the total number of directly dated individuals in this study 38 (45 total dates). Dates were calibrated using OxCal v.4.4[96] with the IntCal20 atmospheric curve[97].

**Liquid chromatography–tandem mass spectrometry and data analysis.** Archaeological dental calculus samples from 45 individuals and 5 extraction non-template controls were processed using a filter-aided sample-preparation protocol, modified for ancient proteins (https://doi.org/10.17504/protocols.io.7vwhn7e). In brief, dental calculus was demineralized in 0.5 M EDTA, and proteins were solubilized and reduced using SDS lysis buffer (4% SDS, 0.1 M DTT, 0.1 M Tris HCl). Buffer exchange in 8 M urea and total protein isolation were performed using a Microcon 30 kDa centrifugal filter unit with an Ultracel-30 membrane (Millipore), followed by alkylation using iodoacetamide. Following buffer replacement with triethylammonium bicarbonate (TEAB; 0.05 M), the proteins were digested overnight with sequencing-grade modified trypsin (Promega) at 37 °C. Peptides were recovered by centrifugation in TEAB and acidified with trifluoroacetic acid to pH <3 and desalted using C18 stage tips (Pierce). Peptides were analysed by liquid chromatography–tandem mass spectrometry using a Q-Exactive mass spectrometer (Thermo Fisher Scientific) coupled to an ACQUITY UPLC M-Class system (Waters AG) at the Functional Genomics Center Zurich of the University/ETH Zurich. Spectra were acquired from 300–1,700 $m/z$ with an automatic gain control target of $3 \times 10^6$, a resolution of 70,000 (at 200 $m/z$) and a maximum injection time of 110 ms. The quadrupole isolated precursor ions with a 2.0 $m/z$ window, a $5 \times 10^4$ automated gain control value and a maximum fill time of 110 ms. Twelve of the most intense precursor ions for each MS$_1$ scan were fragmented via high collision dissociation with a normalized collision energy of 25, scanned with a resolution of 35,000 (at 200 $m/z$) and a fixed first mass of 200 $m/z$. An intensity threshold of $9.1 \times 10^3$ was applied for MS$_2$ selection, and singly charged ions were excluded. Filter criteria for MS$_2$ selection were an intensity threshold of $9.1 \times 10^3$, and unassigned, singly charged ions were excluded. Selected precursor ions were put onto a dynamic exclusion list for 30 s. For liquid chromatography, the solvent composition at the two channels was 0.1% formic acid in water for channel A and 0.1% formic acid in acetonitrile for channel B. Next, 4 µl of each peptide sample was loaded onto a trap column

(Symmetry C18, 100 Å, 5 µm, 180 µm × 20 mm; Waters AG) with a flow rate of 15 µl min⁻¹ of 99% solvent A for 60 s at room temperature. Peptides eluting from the trap column were refocused and separated on a C18 column (HSS T3 C18, 100 Å, 1.8 µm, 75 µm × 250 mm; Waters AG). The column temperature was 50 °C. Peptides were separated over 73 min with the following gradient: 8–22% solvent B in 49 min, 22–32% solvent B in 11 min and 32–95% solvent B in 5 min. The column was cleaned with 95% solvent B for 5 min after the separation and re-equilibrated at loading condition for 8 min before initializing the next run. Potential contamination was monitored using extraction blanks.

Tandem mass spectra were converted to Mascot generic files by MSConvert version 3.0.11781 using the 100 most intense peaks in each spectra. All tandem mass spectrometry samples were analysed using Mascot (Matrix Science, version 2.6.0). Mascot was set up to search the SwissProt Release 2019_08 database (560,823 entries) assuming the digestion enzyme trypsin, with automatic decoy option. Mascot was searched with a fragment ion mass tolerance of 0.050 Da and a parent ion tolerance of 10.0 ppm. The number of missed cleavages was specified as one. Carbamidomethyl of cysteine was specified in Mascot as a fixed modification. Deamidation of asparagine and glutamine and oxidation of methionine and proline were specified in Mascot as variable modifications.

Scaffold version 4.9.0 (Proteome Software Inc.) was used to validate protein and peptide identifications for each sample. Peptide identifications were accepted if they could be established at greater than a 90% probability by the PeptideProphet algorithm. Protein identifications were accepted if they could be established at a greater than 95% probability and contained at least two unique peptides. Probabilities for proteins were assigned using the ProteinProphet algorithm[98]. Proteins that contained similar peptides that could not be differentiated based on tandem mass spectrometry analysis alone were grouped to satisfy the principles of parsimony, and proteins that shared significant peptide evidence were grouped into clusters. Peptide identifications were accepted if they could be established at a greater than 90% probability using the PeptideProphet algorithm[99] with Scaffold delta-mass correction. Individual protein and peptide false discovery rates are listed in Supplementary Data 3.

**Reporting Summary.** Further information on research design is available in the Nature Research Reporting Summary linked to this article.

## Data availability
Raw data files are available through the ProteomeExchange Consortium via the PRIDE partner repository under accession PDX027728. Source data are provided with this paper.

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

## Acknowledgements

This research was funded by the Max Planck Society, the European Research Council under the European Union's Horizon 2020 research and innovation programme (grant agreement numbers 771234-PALEoRIDER, W.H.; 834616-ARCHCAUCASUS, S.H.; 678901-FoodTransforms, A.S.; and 804884-DAIRYCULTURES, C.W.), the ERA.Net RUS Plus initiative (S&T-277-BIOARCCAUCASUS, S.H., A.Bu.), and the Russian Science Foundation (RSF grant 21-18-00026, N.S.).

## Author contributions

C.W., W.H., S.H., E.H., and J.K. designed the study. S.R., S.H., A.R.K, V.E.M., A. Belinsky, N.B., A. Buzhilova, F.G., B.L., P.G., B.J., J.E., E.I., E.H., S.N., K.M., P.O., K.N., N.S., E.K., A.I.K., A.A. Kalmykov, A.A. Khokhlov, R.V.S., R.K., S.V.S., E.S. and M.K. provided materials and resources. A.S. and R.H. performed laboratory experiments. A.S. analysed the data. C.W., A.S., S.R., T.H., S.H. and W.H. assisted with data interpretation. A.S., S.R., T.H., W.H. and C.W. wrote the manuscript with contributions from all co-authors.

## Funding

## Competing interests

The authors declare no competing interests.

## Additional information

**Correspondence and requests for materials** should be addressed to Svend Hansen, Wolfgang Haak or Christina Warinner.

# Reporting Summary

## Statistics

For all statistical analyses, confirm that the following items are present in the figure legend, table legend, main text, or Methods section.

| n/a | Confirmed | |
|---|---|---|
| ☐ | ☒ | The exact sample size (*n*) for each experimental group/condition, given as a discrete number and unit of measurement |
| ☐ | ☒ | A statement on whether measurements were taken from distinct samples or whether the same sample was measured repeatedly |
| ☒ | ☐ | The statistical test(s) used AND whether they are one- or two-sided<br>*Only common tests should be described solely by name; describe more complex techniques in the Methods section.* |
| ☐ | ☒ | A description of all covariates tested |
| ☐ | ☒ | A description of any assumptions or corrections, such as tests of normality and adjustment for multiple comparisons |
| ☐ | ☒ | A full description of the statistical parameters including central tendency (e.g. means) or other basic estimates (e.g. regression coefficient) AND variation (e.g. standard deviation) or associated estimates of uncertainty (e.g. confidence intervals) |
| ☒ | ☐ | For null hypothesis testing, the test statistic (e.g. *F*, *t*, *r*) with confidence intervals, effect sizes, degrees of freedom and *P* value noted<br>*Give P values as exact values whenever suitable.* |
| ☒ | ☐ | For Bayesian analysis, information on the choice of priors and Markov chain Monte Carlo settings |
| ☒ | ☐ | For hierarchical and complex designs, identification of the appropriate level for tests and full reporting of outcomes |
| ☒ | ☐ | Estimates of effect sizes (e.g. Cohen's *d*, Pearson's *r*), indicating how they were calculated |

*Our web collection on statistics for biologists contains articles on many of the points above.*

## Software and code

Policy information about availability of computer code

| | |
|---|---|
| Data collection | Xcaliber (ThermoFisher Scientific) |
| Data analysis | Mascot (Matrix Science, v 2.6.0); MSConvert v 3.0.11781; Scaffold (Proteome Software, v 4.9.0); OxCal v 4.4 with the IntCal20 atmospheric curve. |

For manuscripts utilizing custom algorithms or software that are central to the research but not yet described in published literature, software must be made available to editors and reviewers. We strongly encourage code deposition in a community repository (e.g. GitHub). See the Nature Portfolio guidelines for submitting code & software for further information.

## Data

Policy information about availability of data

All manuscripts must include a data availability statement. This statement should provide the following information, where applicable:
- Accession codes, unique identifiers, or web links for publicly available datasets
- A description of any restrictions on data availability
- For clinical datasets or third party data, please ensure that the statement adheres to our policy

Raw data files are available through the ProteomeExchange Consortium via the PRIDE partner repository under accession PDX027728.

# Field-specific reporting

Please select the one below that is the best fit for your research. If you are not sure, read the appropriate sections before making your selection.

☐ Life sciences ☐ Behavioural & social sciences ☒ Ecological, evolutionary & environmental sciences

For a reference copy of the document with all sections, see nature.com/documents/nr-reporting-summary-flat.pdf

# Ecological, evolutionary & environmental sciences study design

All studies must disclose on these points even when the disclosure is negative.

| | |
|---|---|
| Study description | This study conducted shotgun tandem mass spectrometry on proteins present in human dental calculus obtained from 29 archaeological sites in Russia and Azerbaijan. Proteins were identified using Mascot and validated using Scaffold, and dietary proteins were analyzed to reconstruct past dairy pastoralism strategies. |
| Research sample | 46 human dental calculus specimens were analyzed from 45 individuals at 29 archaeological sites. A comprehensive overview of these specimens is provided in Dataset S1, and detailed osteological and archaeological context data is provided in the SI. |
| Sampling strategy | We sought to comprehensively sample archaeological sites relevant for the research question. Dental calculus was collected from individuals for whom at least 5 mg of dental calculus was available, but care was taken to avoid sampling individuals with small amounts of calculus, for whom this collection would represent sample exhaustion. |
| Data collection | Proteomics data was collected using a Q-Exactive mass spectrometer (Thermo Scientific, Bremen, Germany) coupled to an ACQUITY UPLC M-Class system (Waters AG, Baden-Dättwil, Switzerland) at the Functional Genomics Center Zurich of the University/ETH Zurich. AMS radiocarbon dating was performed at the Curt-Engelhorn-Zentrum Archäometrie (CEZA) in Mannheim, Germany, the Finnish Museum of Natural History (Hela) in Helsinki, Finland, the Oxford Radiocarbon Accelerator Unit (OxA) in Oxford, UK, the Kiev Radiocarbon Laboratory (KI) in Kiev, Ukraine, Russian Academy of Sciences (GIN) in Moscow, Russia. |
| Timing and spatial scale | Samples were analyzed over the period 2016-2021. |
| Data exclusions | Data generated during a QC-failed instrument run (failure to achieve chromatographic separation on the UPLC prior to analysis) were discarded and the experiment repeated. One collected sample was excluded from the study after a labeling error was discovered that made it its archaeological provenance uncertain. |
| Reproducibility | Sample ZO002 was analyzed twice, yielding highly similar results. |
| Randomization | NA |
| Blinding | NA |

Did the study involve field work? ☐ Yes ☒ No

# Reporting for specific materials, systems and methods

We require information from authors about some types of materials, experimental systems and methods used in many studies. Here, indicate whether each material, system or method listed is relevant to your study. If you are not sure if a list item applies to your research, read the appropriate section before selecting a response.

## Materials & experimental systems

| n/a | Involved in the study |
|---|---|
| ☒ | ☐ Antibodies |
| ☒ | ☐ Eukaryotic cell lines |
| ☐ | ☒ Palaeontology and archaeology |
| ☒ | ☐ Animals and other organisms |
| ☒ | ☐ Human research participants |
| ☒ | ☐ Clinical data |
| ☒ | ☐ Dual use research of concern |

## Methods

| n/a | Involved in the study |
|---|---|
| ☒ | ☐ ChIP-seq |
| ☒ | ☐ Flow cytometry |
| ☒ | ☐ MRI-based neuroimaging |

# Palaeontology and Archaeology

| | |
|---|---|
| Specimen provenance | Various archaeological sites in Russia and Azerbaijan; see Data S1 and the SI for a comprehensive list of archaeological sites, locations, and permit license numbers. |
| Specimen deposition | Contact information for each specimen is provided in the SI. |

| Dating methods | New radiocarbon dates and lab codes are provided in Data S1. |

☒ Tick this box to confirm that the raw and calibrated dates are available in the paper or in Supplementary Information.

| Ethics oversight | Study consists of the analysis of previously excavated archaeological remains, which are exempt from human subjects ethical review; excavation partners providing specimens are coauthors on the study and excavation licenses are provided in the SI. |

Note that full information on the approval of the study protocol must also be provided in the manuscript.

