## [Peer Review File · Nature Ecology & Evolution]

Peer Review Information

Journal: Nature Ecology & Evolution

Manuscript Title: Emergence and intensification of dairying in the Caucasus and Eurasian steppes

Corresponding author name(s): Svend Hansen, Wolfgang Haak, Christina Warinner

Editorial Notes:

Reviewer Comments & Decisions:

Decision Letter, initial version:
--

15th October 2021

Dear Christina,

Your manuscript entitled "Emergence and intensification of dairying in the Caucasus and Eurasian steppes" has now been seen by four reviewers, whose comments are attached. The reviewers have raised a number of concerns which will need to be addressed before we can offer publication in Nature Ecology & Evolution. We will therefore need to see your responses to the criticisms raised and to some editorial concerns, along with a revised manuscript, before we can reach a final decision regarding publication.

I just want to draw your attention to one of the reviewer comments that the raw data were not available to access without compromising anonymity--please could you look into this for when you submit the revision.

We therefore invite you to revise your manuscript taking into account all reviewer and editor comments. Please highlight all changes in the manuscript text file [OPTIONAL: in Microsoft Word format].

* If you have not done so already please begin to revise your manuscript so that it conforms to our Article format instructions at <http://www.nature.com/natecolevol/info/final-submission>. Refer also to any guidelines provided in this letter.

[REDACTED]

Nature Ecology & Evolution is committed to improving transparency in authorship. As part of our efforts in this direction, we are now requesting that all authors identified as 'corresponding author' on published papers create and link their Open Researcher and Contributor Identifier (ORCID) with their account on the Manuscript Tracking System (MTS), prior to acceptance. ORCID helps the scientific community achieve unambiguous attribution of all scholarly contributions. You can create and link your ORCID from the home page of the MTS by clicking on 'Modify my Springer Nature account'. For more information please visit www.springernature.com/orcid.

[REDACTED]

Reviewer expertise:

Reviewer #1: proteomics

Reviewer #2: zooarchaeology of the steppe region

Reviewer #3: proteomics

Reviewer #4: zooarchaeology of the steppe region

Reviewers' comments:

Reviewer #1 (Remarks to the Author):

The human activity including milk consumption in the Pontic-Caspian Steppe and neighboring South Caucasus, Oka-Volga-Don, and East Urals region had a great impact or reflected the development of pastoralism in Eurasia steppe. Although some previous research discussed the milking in the Pontic-Caspian region, when the milking began and how the milking changed in the Pontic-Caspian Steppe and neighboring area still need more work.

This study analyzed more human dental calculus sample across different periods and areas, compared to previous work, and identified the species of milk by using proteomics. In particular, this study provides new milking evidence in Eneolithic and Early Iron Age, which reflect ancient human immigration and the impact of climate change. This work will promote further understanding the culture evolution during this area. However, the study needs some revision.

1)The authors should incorporate the results of previous research, "Dairying enabled Early Bronze Age Yamnaya steppe"

So the authors could use more data to better describe the milking change in the wide research area.

2)It seems that this study overthrows some conclusions in previous work, such as the absence of milking in Eneolithic North Caucasus. This point needs more explanation.

3)It's better to give a climate curve to link the milking change and climate change.

4)The milk consumption is diversified between Early Bronze Age and Middle Bronze Age, was there some change of social structure or settlement patterns, society complexity?

Reviewer #2 (Remarks to the Author):

This paper presents convincing data that provide a very strong narrative for the development of dairying and pastoralism in a critical region for both both pastoralist societies and horse domestication that is also linked to a number of major genetic diaspora. This paper is much stronger than the recent Wilken et al paper, that covered similar ground, in terms of its sampling, rigour of dating and quality of discussion. It is well conducted and important work deserving publication with a limited number of minor revisions:

1) Lines 96-101: This section discusses the importance of the Yamnaya genetic diaspora and influence on CWC groups. Nothing said is incorrect. However, this is said in the context of being one of the reasons why their study is important. Perhaps more should be said about how their results contribute to understanding the mechanisms of this spread. Others have implied milking facilitated the spread, but this paper shows earlier milking in the region thus removing such a clear horizon. Indeed, milking was already well established across Europe well before Yamnaya date, in any case. Introduction of LP

doesn't stand up to scrutiny either with limited (and only imputed?) LP in Yamnaya and earlier examples further west now. So perhaps it isn't milk use per se, but maybe simply the mobile economy versus sedentary late Neolithic people with some evidence for increasing crop failure? Clearly this point would need to be returned to and elaborated upon in the discussion section.

2) 121-122: On horse controversy, when using citation 42, best to simultaneously cite original paper, already later cited (80) and perhaps the rebuttal to 42 to allow the readers to properly judge this debate.

3) 142-145: I think this summary of the regional isotopic work in this paper shows the general strength of discussion. So many people (who should pay better attention) regularly misinterpret isotopic data for pastoralists, particularly in relation to $\delta^{15}N$ values. Such misunderstanding has, in recent cases, led to inappropriate reservoir adjustments to C14 dates. This paper gets this right.

4) 341: When discussing Botai milking evidence you really need note that only 2 samples were ever studied for protein in calculus, whilst many more sherds were studied by lipid residue analysis. Using the evidence from those two samples in reference 7 may suit your argument, but you know from that sampling the result is not sound, so should not hide that for the convenience of your argument. Indeed, reference 7 fails to find convincing evidence milk in 11 Pontic-Caspian Eneolithic samples, which contradicts the results from this current paper. So it is somewhat disingenuous to make full use of a very limited 2 negative samples when it suits you, but ignore 11 when it doesn't! The additional nuance does not weaken your paper. Just makes it more rigorous and a better representation of the facts.

342: earliest direct evidence is only first proteomic evidence

4) The discussion of horse milking misses an important point. Your sequence deals only with mixed pastoralists with access to ruminant food products. In the spread to the East this involved total population replacement after the MBA, and this was total replacement of people and horses (with DOM2). Earlier studies using zooarchaeological data and lipid residue analysis also conclude that the majority of food supply came from ruminants and horse meat consumption was quite rare apart from in ritual contexts. The lipid residue analyses also detected no horse milking during the BA. However, the period before that involved different human populations intensively exploiting a different horse lineage without access to ruminant milk, but likely awareness of other's milking behaviours in neighbouring contemporary regions. So this paper slightly overstretches its conclusion because it presents a single narrative development in a single sequence, when in fact the earlier sequence in the region was completely different and utterly cut off. Indeed, this paper presents no new data on that part of the sequence. One is left with some evidence for horse milking from lipid residues challenged by only two proteomic samples in another study. I recommend noting the lack of continuity of sequence in the Central Steppe and the ambiguity of current evidence in the earlier part of it.

All the above are minor but important nuances to an otherwise brilliant paper. However, I think these minor adjustments would increase rigour and present an excellent platform for future research.

Reviewer #3 (Remarks to the Author):

Dear authors,

It was a pleasure reading your manuscript, but I do have some points of revision before I can

recommend publication.

I find that your results will provide valuable and novel details within the specific research area and I do not question the presented data or your conclusions. As you will read from my review, I am not an archaeologist and I will focus my comments on the proteomics part and a more general view.

General:

The manuscript does not come out very trustworthy when the phrasing proteome is used and then consequently only one or two milk proteins are described. I believe that proteomes are more than just one or two proteins. At the same time I find it hard to believe that only milk proteins were identified considering that all entries in Uniprot were searched. Do you have another paper in mind for the rest of the data?

It would immensely benefit the manuscript if either, every identified protein were included and described as proteome or that the focus were exclusively on milk proteins, which would require a new Mascot search only on milk proteins. If the latter, use phrasing as 'mass spectrometry analysis of milk proteins' not proteome analysis or proteomics.

I have not received any reviewer login to check raw data on PRIDE, so I cannot comment on that or the search output.

Specific:

P2 L68; use of proteomes

P3 L106-108; Specify difference between 'wild boar (*Sus scrofa*)' and 'pigs (*Sus scrofa*)?'. Proteomics-wise, it would be the same.

P4 L163; 'in the North Caucasus, with milk proteins identified in 25 of 26 tested individuals'. In L157 same page you have n=27. 25 of 27?

P4 L182; 'All dental calculus samples yielded proteomes consistent with an oral microbiome profile, and age-associated N/Q deamidation was a top modification across the dataset (Dataset S2)?'. I see no data in S2 indicating microbiome or proteome... only milk proteins.

P4 L184; 'No dietary proteins were detected in non-template extraction controls or injection blanks (Dataset S2)'. Please describe the difference between the two and the procedure. I only find info on 5 blanks in dataset S2.

P9/10; The LC-MS/MS and Data Analysis section. Since the whole paper is based on mass spectrometry data it would be highly recommended to share the details on the specific MS method. Such as full scan resolution, fragment scan resolution, topN method, spray voltage, which column material and length were used for separation of peptides, how long and steep gradient... Basic proteomic paper parameters.

I wish you all the best.

Rosa R. Jersie-Christensen

Reviewer #4 (Remarks to the Author):

This paper, which analyzes human dental calculus collected from archaeological specimens from the North Caucasus and nearby regions, represents an important and interesting set of data that sheds some light on the consumption of dairy products from domesticated herd animals. The research is

aimed at answering an important question in the archaeology of the region and the results of the work will be of interest to the field. However, I cannot recommend the manuscript for publication in its current form.

The primary issue with the article, as currently written, is that it over-interprets the data that the authors have generated. The interpretations that they propose ignore some of the basic deficiencies, or more perhaps optimistically, challenges of their samples. The authors have tested dental calculus from 45 people across 29 sites from a time period spanning ~6,000 years. Needless to say, this is thin coverage – both temporally and spatially. The sample of 27 people from sites in the North Caucasus is closer to being sufficient, but the samples from the South Caucasus, Oka-Volga-Don, and eastern Urals are too small to support interpretations beyond random positive evidence of dairy consumption. Moreover, within the regional datasets, they have sampled only a small number of individuals from the same site (some from different phases). Thus, the dataset they've gathered cannot meaningfully address the possibility that there are dietary differences within communities.

The authors have failed to directly state and support certain assumptions that appear to be at the foundations of their interpretations. It seems that they are working from the assumption that if there is dairying (be it consumption or a dairying "economy"), then everyone (or a very large percentage of people) would be consuming dairy products in such a way that dairy products would be detectable in the calculus. I am not sure this is a reasonable assumption, given that a recent paper on the method indicates that: "Little is currently known about how dietary proteins become trapped within dental calculus, and variation in this process may influence downstream protein recovery and identification success. Until we understand the degradation of these proteins, we cannot conclude that the absence of evidence is the evidence of absence." (Hendy et al. 2018, 5).

Otherwise, how are we to understand what conclusions to draw when a single individual from their dataset does not show evidence of dairy consumption in their calculus? On one hand, if this individual is truly from a community that did not consume dairy, it is truly evidence of an absence of dairy consumption. But if we are dealing with a community where only 1 in 4 people consume dairy in a way where we would expect to be able to detect it in the dental calculus, then it isn't truly evidence of an absence of dairying. It is merely an artifact of choosing too small of sample to measure the prevalence of the event/trait in the population. This is true at the level of site and the level of the region and/or time period.

Likewise, the authors need to be clear about what level of consumption constitutes the phenomenon of interest. Does dairying only matter when it is everyone in the community? 75%? 50%? 25%? Any consumption of milk whatsoever? The current experimental design is only adequate to one scenario – a very high-level of consistent dairy consumption. Connected to this issue is the slippage in how the authors use the term "dairying", which admittedly, is a very common feature of the literature on prehistoric dairying. The authors should think carefully about what they mean when they say dairying. Is it merely any human consumption of nonhuman milk products? Or are they referring to a particular kind of pastoralist economy (with the attendant assumption that milk products would provide a greater portion of the diet or have a more important role in the pastoralist economy). The ability or inability of the presently-available data and methods to address either form of dairying needs to be explicitly addressed.

To that end, I've identified some specific instances in the manuscript where these issues have led to problems:

Lines 277-278:

The authors write that “Surprisingly, however, the dairy economy remained exclusively focused on sheep.” This claim is made in reference to the data the authors have collected from 7 individuals across 1,000 years. This is an overinterpretation, since the dataset can only positively identify presence, especially without any detailed zooarchaeological data from these sites to support this conclusion.

Lines 298-299:

The authors state that “Most individuals of the Middle Bronze Age Catacomb, NCC, Late NCC, and Lola cultures consumed the dairy products of two or three livestock species.” As written, in the larger context of the article, this statement is misleading (and is likely to be misunderstood). More adequate would be something like: “Most of the individuals from the Middle Bronze Age Catacomb, NCC, Late NCC, and Lola cultures that were included in the study [alternate version: that were tested] consumed dairy products from two or three livestock species.”

Lines 311-313:

The authors write ““Our results suggest that during the Lola period, water-demanding cattle began to decrease in economic importance from the preceding Catacomb and NCC periods, as only one out of six Lola individuals yielded evidence for cattle milk consumption.” This claim is not supported by the evidence. Cattle could have been used for non-milk purposes, which is precisely what the authors suggested previously in the paper when they interpreted the Maykop data. Without any corroboration from zooarchaeological data, the climate explanation cannot be given precedence.

Lines 377-380:

The claims the authors make here are not tenable because there cannot be a legitimate comparison between the datasets they present from the North and South Caucasus. The data from the North and South Caucasus are divergent in terms of their sample sizes and in terms of the adequacy of regional coverage.

Line 388:

The authors write about ““the near-complete lack of evidence for ruminant milk consumption from the seven individuals” that they tested from the Oka-Volga-Don region. This line could be re-written, and would perhaps be more accurate, as “14% of individuals tested showed evidence of ruminant milk consumption.”

Lines 409-411:

The authors write “Dairying was integral for the spread of animal husbandry by groups crossing the Caucasus mountains from south to north during the Eneolithic”. The evidence presented in the paper is not able to support this claim. Not only is the data from the South Caucasus highly partial, it is also basically contemporaneous with the data they’ve collected for the North.

Lines 413-414:

The authors write "Initial pastoralist strategies were specialized, with dairying focused on sheep and cattle used for traction". This overstates the evidence. The claim might be applicable if limited to the Maykop, though that is also probably still an overreach.

Figure 3:

This figure is very misleading and probably should be abandoned. It does not show what is missing from the datasets (in terms of temporal and regional coverage), producing an incorrect impression. Particularly troubling are: 1) the lack of any data from Georgia or Armenia, 2) the absence of any data from anywhere in the South Caucasus between 3500-2500 BC, 3) the absence of data from the North Caucasus prior to ~ 4500 BC, and 4) the fact that there is only data for the E. Urals for one period. Overall, the figure presents a misleading picture of the available data and is highly likely to be misinterpreted. It might be possible to make a version of this figure that only focuses on the North Caucasus, since that region has more adequate coverage.

Finally, my reading of the manuscript has also left me with some more specific methodological questions that ideally would be addressed somewhere (possibly in supplementary materials, since word counts are often quite limiting).

- 1) Is all of the available calculus sampled? Or is there a distinct procedure for sampling a portion of the available calculus?
- 2) Are the authors able to suggest how much time is averaged in the sample calculus? What factors might influence inter-individual variation in the amount of time represented by the sampled calculus?
- 3) If the calculus sampled represents a somewhat shorter process of accumulation and averaging, how might any seasonality in milk consumption or the possibility that people of different ages consumed more or less dairy products impact the interpretation of data produced by this method?

Works Cited:

Hendy, Jessica, Christina Warinner, Abigail Bouwman, Matthew J. Collins, Sarah Fiddyment, Roman Fischer, Richard Hagan, et al. 2018. "Proteomic Evidence of Dietary Sources in Ancient Dental Calculus." *Proceedings of the Royal Society B: Biological Sciences* 285 (1883): 20180977. <https://doi.org/10.1098/rspb.2018.0977>.

*****END*****

Author Rebuttal to Initial comments

Point-by-Point Response to Reviewer Comments for NATECOLEVOL-210914661**“Emergence and intensification of dairying in the Caucasus and Eurasian steppes”**

We received 30 specific comments from four reviewers. We present our responses to each comment below. Changes made are marked in blue in the revised manuscript files.

Reviewer 1:

The human activity including milk consumption in the Pontic-Caspian Steppe and neighboring South Caucasus, Oka-Volga-Don, and East Urals region had a great impact or reflected the development of pastoralism in Eurasia steppe. Although some previous research discussed the milking in the Pontic-Caspian region, when the milking began and how the milking changed in the Pontic-Caspian Steppe and neighboring area still need more work. This study analyzed more human dental calculus sample across different periods and areas, compared to previous work, and identified the species of milk by using proteomics. In particular, this study provides new milking evidence in Eneolithic and Early Iron Age, which reflect ancient human immigration and the impact of climate change. This work will promote further understanding the culture evolution during this area. However, the study needs some revision.

Comment 1: The authors should incorporate the results of previous research, “Dairying enabled Early Bronze Age Yamnaya steppe” So the authors could use more data to better describe the milking change in the wide research area.

We attempted to do this, but in downloading and reanalyzing the data in Wilkin et al. 2021, we identified multiple inconsistencies between the data reported in the main text, SI, and files uploaded to the PRIDE repository. These include missing data files, additional data files not reported in the study, methodological discrepancies, and inconsistencies in the results reported in the study’s main figures and supplementary tables. Due to these discrepancies and missing files, we were unable to unambiguously reconstruct the findings of the published paper or to reanalyze the data ourselves. Consequently, we are unable to complete the reviewer’s request.

Comment 2: It seems that this study overthrows some conclusions in previous work, such as the absence of milking in Eneolithic North Caucasus. This point needs more explanation.

With the exception of two Early Bronze Age samples, all Eneolithic and Early Bronze Age individuals analyzed in the Wilkin et al. 2021 study fall from outside the North Caucasus region in an area to the northeast that spans the Don-Volga and Trans-Ural regions. Among their Eneolithic individuals, they do not find evidence of dairying at the sites

of Khvalynsk and Khlopkov Bugor, which are located in the Upper Volga River valley. Since our study has a strong focus on the North Caucasus region, where we argue regional dairying first begins, we do not find these results conflicting. Rather, we believe our results establish the first clear picture of ancient dairying north of the Caucasus mountains and establish the regional and temporal framework by which it first spreads into the northern piedmont slopes of the Caucasus during the 5th millennium BCE Eneolithic and then subsequently spreads northwards into the steppes during the Early Bronze Age.

Regarding horse milking, we find it difficult to follow the underlying chronological data from the Wilkin et al. 2021 study, as several of the dates have been adjusted for reservoir effects, but the reporting of this correction is inconsistent in the SI and SI tables. Their “Early Bronze Age” individuals from Krivyanskiy IX have modelled radiocarbon date ranges in excess of 1000 years (e.g., KRI9 K.4 N-21 - 3341-2203 calBCE), which severely impedes understanding what time period the proteomic data represent. Taken together, it is difficult to accurately and precisely place the two Krivyanskiy IX dental calculus samples with evidence of horse milk within chronological time.

Even if the two Krivyanskiy IX individuals cannot be precisely dated, it is possible that there was sporadic horse milking during the third millennium BCE, as a recent paleogenomic study by Librado et al. 2021 (<https://www.nature.com/articles/s41586-021-04018-9>) showed that the horse lineage that characterizes modern horses emerged in the Volga-Don-North Caucasus region as early as the 4th millennium BCE and began spreading widely by the end of the 3rd millennium BC and start of the 2nd millennium BCE. Since the Krivyanskiy IX individuals represent a single find, and our denser sampling of the North Caucasus did not detect any other evidence of horse milk proteins until the 9th century BCE, we updated the text to note that - to the degree that early horse milking was practiced in the region - it was likely sporadic and limited, even if emerging first out of traditions of horse domestication in the lower Volga-Don region. We adjusted the sentence in the Discussion section to reflect that the likely region and time period of horse domestication is now known, and we also added clarifying points about the limited evidence for horse milking at the end of that paragraph.

Our research helps to fill a major gap in the archaeology of the North Caucasus, where there has remained a lack of systematic zooarchaeological data (e.g., faunal age and sex distributions) to identify and characterize the dairy economies of Eneolithic and Early Bronze Age communities. Our proteomic data confirms earlier inferences made using stable isotope analysis and provides direct evidence of the Eneolithic establishment and further Bronze Age development of dairying economies in the North Caucasus. We believe that our proteomic data advances archaeological knowledge by definitively showing the rise of dairying and its evolution in this important region for the spread of people, technologies, and traditions throughout Eurasia.

Comment 3: It's better to give a climate curve to link the milking change and climate change.

We have also considered this, but the challenge here is that there is no one climate curve, let alone an adequate single proxy for the steppe and forest steppe zones under study here. The most extensive study of North Caucasus climate shifts over this period is documented by paleosols and is reported in Shishlina (2008:221-223). Regional climatic fluctuation and aridification is further documented in floral and faunal stable isotope values (Knipper et al. 2020). Further evidence for a climatic deterioration around 4.2 ka cal BP comes from a number of proxies in the northern hemisphere ranging from $\delta^{18}\text{O}$ isotope data from Greenland ice cores or stalagmites in caves (western China) to lake sediments from various regions in Iceland, the Mediterranean, and sea level changes of the Dead Sea (see e.g. Jung & Weninger 2014). We have ensured that these sources are cited when ecological changes and aridification are mentioned in the text.

Comment 4: The milk consumption is diversified between Early Bronze Age and Middle Bronze Age, was there some change of social structure or settlement patterns, society complexity?

The shift to a more diversified dairy strategy occurs during the Yamnaya period, and we first observe it in a Late Yamaya individual. This pattern is also found during subsequent periods and among later cultural groups in the North Caucasus. Broadly, these later MBA groups, which include the North Caucasian Culture (NCC), Catacomb, and Lola, are more mobile and exhibit reduced social hierarchy compared to earlier EBA groups, such as the Maykop and Steppe Maykop (Reinhold et al. 2019). The MBA groups were mobile pastoralists and did not have large or permanent settlements. Because the region lacks archaeological data from settlements, our approach of recovering subsistence information directly from human skeletal remains is crucial for understanding social behaviors linked to food production. Unfortunately, the limited nature of the archaeological data for the region, which is primarily confined to mortuary contexts, does not allow us to make further conclusions about shifts in social structure or complexity.

*Reinhold, Sabine. "The Maykop legacy – new social practice and new technologies in the 4th millennium BC in the North Caucasus." In: Sławomir Kadrow, Johannes Müller (eds.), *Habitus?: the social dimension of technology and transformation*. Leiden: Sidestone Press, 2019, 87-113*

Reviewer 2:

This paper presents convincing data that provide a very strong narrative for the development of dairying and pastoralism in a critical region for both pastoralist societies and horse domestication that is also linked to a number of major genetic diaspora. This paper is much stronger than the recent Wilken et al paper, that covered similar ground, in terms of its sampling, rigour of dating and quality of discussion. It is well conducted and important work deserving publication with a limited number of minor revisions:

Comment 5: Lines 96-101: This section discusses the importance of the Yamnaya genetic diaspora and influence on CWC groups. Nothing said is incorrect. However, this is said in the context of being one of the reasons why their study is important. Perhaps more should be said about how their results contribute to understanding the mechanisms of this spread. Others have implied milking facilitated the spread, but this paper shows earlier milking in the region thus removing such a clear horizon. Indeed, milking was already well established across Europe well

before Yamnaya date, in any case. Introduction of LP doesn't stand up to scrutiny either with limited (and only imputed?) LP in Yamnaya and earlier examples further west now. So perhaps it isn't milk use per se, but maybe simply the mobile economy versus sedentary late Neolithic people with some evidence for increasing crop failure? Clearly this point would need to be returned to and elaborated upon in the discussion section.

Although dairying and a mobile economy appear to have been key components of the Yamnaya expansion, we agree with the reviewer that their emergence predates the Yamnaya and thus their invention alone did not trigger the Yamnaya expansions. As we show in this study, dairying had been practiced in the piedmont zone of the North Caucasus since at least 4200 BCE and on the steppe since at least 3700 BCE, and mobile pastoralism was first developed during the Maykop period. Thus, mobile dairy pastoralism was already a feature of steppe life in the region prior to the Yamnaya. However, the capacity for mobility and a subsistence strategy compatible with high mobility (i.e., a 'walking larder' (Clutton-Brock 1989)), would have been necessary to sustain the Yamnaya expansion in the manner as it occurred. Therefore, it is not correct for Wilkin et al. (2021) to advance a narrative linking the initial emergence of steppe dairying with Yamnaya expansions. Since this is a timely topic, we added a short sentence at the beginning of the Discussion subsection about the diversification of dairying to highlight the disconnect between earlier dairying and the Yamnaya expansions, so readers are aware of the discrepancy. We also added a short clause at the end of the first paragraph in the Discussion to further highlight that our dataset shows the emergence of dairying well before the Yamnaya horizon.

Comment 6: Lines 121-122: On horse controversy, when using citation 42, best to simultaneously cite original paper, already later cited (80) and perhaps the rebuttal to 42 to allow the readers to properly judge this debate.

We have replaced citation 42 with the following reference:

Librado, P., Khan, N., Fages, A., Kusliy, M.A., Suchan, T., Tonasso-Calvière, L., Schiavinato, S., Alioglu, D., Fromentier, A., Perdereau, A. and Aury, J.M., 2021. The origins and spread of domestic horses from the Western Eurasian steppes. *Nature*, pp.1-7.

We do not wish to dwell on the Botai management/domestication debate, since it is outside the analytical and regional scope of this paper. We clarified that Botai horses reflect the DOM1 lineage, while the horses from the Volga-Don North Caucasus region reflect the DOM2 lineage and its ancestors.

Comment 7: Lines 142-145: I think this summary of the regional isotopic work in this paper shows the general strength of discussion. So many people (who should pay better attention) regularly misinterpret isotopic data for pastoralists, particularly in relation to $\delta^{15}\text{N}$ values. Such misunderstanding has, in recent cases, led to inappropriate reservoir adjustments to C^{14} dates. This paper gets this right.

We thank the reviewer for their positive assessment of our analysis.

Comment 8: Line 341: When discussing Botai milking evidence you really need note that only 2 samples were ever studied for protein in calculus, whilst many more sherds were studied by lipid residue analysis. Using the evidence from those two samples in reference 7 may suit your argument, but you know from that sampling the result is not sound, so should not hide that for the convenience of your argument. Indeed, reference 7 fails to find convincing evidence milk in 11 Pontic-Caspian Eneolithic samples, which contradicts the results from this current paper. So it is somewhat disingenuous to make full use of a very limited 2 negative samples when it suits you, but ignore 11 when it doesn't! The additional nuance does not weaken your paper. Just makes it more rigorous and a better representation of the facts. Line 342: earliest direct evidence is only first proteomic evidence.

Thank you for this suggestion. We have now more clearly cited the evidence for horse milk lipids in Botai pottery. Upon further investigation, we also discovered that Wilkin et al. 2021 did not make their raw data available for the Botai calculus, and so we cannot evaluate or confirm their findings. Consequently, we have removed our statement about the Botai calculus results. Regarding the statement about the “earliest direct evidence”, we have revised this sentence to now state the “earliest proteomic evidence”.

Comment 9: The discussion of horse milking misses an important point. Your sequence deals only with mixed pastoralists with access to ruminant food products. In the spread to the East this involved total population replacement after the MBA, and this was total replacement of people and horses (with DOM2). Earlier studies using zooarchaeological data and lipid residue analysis also conclude that the majority of food supply came from ruminants and horse meat consumption was quite rare apart from in ritual contexts. The lipid residue analyses also detected no horse milking during the BA. However, the period before that involved different human populations intensively exploiting a different horse lineage without access to ruminant milk, but likely awareness of other's milking behaviours in neighbouring contemporary regions. So this paper slightly overstretches its conclusion because it presents a single narrative development in a single sequence, when in fact the earlier sequence in the region was completely different and utterly cut off. Indeed, this paper presents no new data on that part of the sequence. One is left with some evidence for horse milking from lipid residues challenged by only two proteomic samples in another study. I recommend noting the lack of continuity of sequence in the Central Steppe and the ambiguity of current evidence in the earlier part of it. All the above are minor but important nuances to an otherwise brilliant paper. However, I think these minor adjustments would increase rigour and present an excellent platform for future research.

We have revised our section on horse milking to more clearly delineate the evidence for the management and milking of DOM1 and DOM2 equid lineages in the 4th millennium BCE, and we have revised the text to no longer

imply continuity or a single narrative of development. We have also indicated that the modeled dates for the Kriviyansky IX individuals are problematic and difficult to place in chronological time.

Reviewer 3:

It was a pleasure reading your manuscript, but I do have some points of revision before I can recommend publication. I find that your results will provide valuable and novel details within the specific research area and I do not question the presented data or your conclusions. As you will read from my review, I am not an archaeologist and I will focus my comments on the proteomics part and a more general view.

Comment 10: The manuscript does not come out very trustworthy when the phrasing proteome is used and then consequently only one or two milk proteins are described. I believe that proteomes are more than just one or two proteins. At the same time I find it hard to believe that only milk proteins were identified considering that all entries in Uniprot were searched. Do you have another paper in mind for the rest of the data?

This appears to be a simple misunderstanding. We generated proteome-scale data, and have made our raw dental calculus proteome data fully available in the PRIDE repository. Our data acquisition was not targeted, so the data provided in these files represent the dental calculus proteome. We have now reviewed each instance of our use of the term proteome in our manuscript and have ensured that it meets the EMBL-EBI definition of a proteome: “A proteome is a set of proteins produced in an organism, system, or biological context.” Because this study focuses on the prehistory of dairy (rather than the oral microbiome) we have focused our data analysis and interpretation on dietary proteins present within the dental calculus proteome.

The database we used for the generation of protein identifications was the curated, non-redundant SwissProt database. We selected this database because it has good representation of ruminant and equid milk proteins, while also including a broad and representative range of proteins from both eukaryotic and bacterial taxa.

Using a full mass scan approach, we identified a high prevalence and abundance of dental calculus PSMs assigned to milk proteins. These results were expected given the strong archaeological evidence for pastoralism in the region. The remaining data consisted of expected bacterial and human proteins consistent with those previously identified in both modern and ancient dental calculus. We have made our mzid files available in our PRIDE upload so that the non-dietary components of the dental calculus proteome are more easily explored and interrogated.

Comment 11: It would immensely benefit the manuscript if either, every identified protein were included and described as proteome or that the focus were exclusively on milk proteins, which would require a new Mascot search only on milk proteins. If the latter, use phrasing as ‘mass spectrometry analysis of milk proteins’ not

proteome analysis or proteomics. I have not received any reviewer login to check raw data on PRIDE, so I cannot comment on that or the search output.

Fully characterizing the >1000 oral microbiome and human proteins present within dental calculus proteomes is beyond the scope of the current study, which focuses on evidence for milk and dairy product consumption. However, we agree that making this data available for others to analyze is critical. We have provided our raw data (.raw, .mgf, and .mzid files) of identified proteins in the PRIDE data repository, which is now public under the accession PXD027728.

- Project Webpage: <http://www.ebi.ac.uk/pride/archive/projects/PXD027728>
- FTP Download: <ftp://ftp.pride.ebi.ac.uk/pride/data/archive/2021/11/PXD027728>

We disagree that searching our data against a smaller database containing only milk proteins would improve our study. Such an approach may increase the identification of low scoring milk peptides, which would otherwise not pass false discovery rate thresholds with a larger database. Performing highly restrictive database searches, such as on a milk-only protein database, when the proteome complexity is known to be high can lead to misleading results and inaccurate measures of false discovery rate (Knudsen and Chalkley 2011).

Knudsen, Giselle M., and Robert J. Chalkley. 2011. "The Effect of Using an Inappropriate Protein Database for Proteomic Data Analysis." PLoS One 6 (6): e20873

Comment 12: P2 L68; use of proteomes

See our response above to Comment 10.

Comment 13: P3 L106-108; Specify difference between 'wild boar (*Sus scrofa*)' and 'pigs (*Sus scrofa*)'? . Proteomics-wise, it would be the same.

This list of taxa was determined by a previous study using zooarchaeological and morphological techniques that are capable of distinguishing wild and domesticated *Sus* species. We provide the corresponding citations in the text. We did not apply proteomics to distinguish these taxa.

Comment 14: P4 L163; 'in the North Caucasus, with milk proteins identified in 25 of 26 tested individuals'. In L157 same page you have n=27. 25 of 27?

We have corrected the relevant sections of the text. We analyzed a total of 27 individuals from the North Caucasus, of which 26 were milk positive. In total, we analyzed 28 dental calculus samples from the North Caucasus because the dental calculus of individual ZO2002 was analyzed twice (ZO2002.B and ZO2002.D).

Comment 15: P4 L182; 'All dental calculus samples yielded proteomes consistent with an oral microbiome profile, and age- associated N/Q deamidation was a top modification across the dataset (Dataset S2)'?. I see no data in S2 indicating microbiome or proteome... only milk proteins.

This data was originally included as a separate tab within Dataset S2, but we recognize that this made it difficult to find. We have now moved this data into its own spreadsheet (Dataset 2) and have moved the milk PSM summaries to a new file, Dataset S3. Within the new Dataset 2, the top 10 modifications observed for each sample across all proteins (the full dental calculus proteome) are provided.

Comment 16: P4 L184; 'No dietary proteins were detected in non-template extraction controls or injection blanks (Dataset S2)'. Please describe the difference between the two and the procedure. I only find info on 5 blanks in dataset S2.

We have updated the phrasing in both the text and the new Dataset S3 (formerly Dataset S2) to be clearer. A non-template extraction control is a control sample of purified water that was processed in the laboratory alongside samples to detect potential laboratory contamination during protein extraction, digestion, and purification steps. In this study, we generated 5 non-template extraction controls, which are reported in Dataset 3 and have been uploaded to the PRIDE repository. An injection blank is a solvent injection performed between samples during LC-MS/MS analysis. The main purpose of an injection blank is to flush the LC column between samples. Although injection blanks are run using different ramp parameters and an abbreviated analysis time in order to force remaining peptides off the column, it is possible to analyze peaks generated by injection blanks to identify potential peptide carry-over between samples. We performed this analysis and did not detect any dietary PSMs in the injection blanks. We did not upload the injection blank files to PRIDE because they were generated and analyzed differently than both the samples and non-template extraction controls, and thus they are not directly comparable; rather we analyzed them strictly as an internal quality control measure. We have expanded the methods section to make these steps clearer.

Comment 17: P9/10; The LC-MS/MS and Data Analysis section. Since the whole paper is based on mass spectrometry data it would be highly recommended to share the details on the specific MS method. Such as full scan resolution, fragment scan resolution, topN method, spray voltage, which column material and length were used for separation of peptides, how long and steep gradient... Basic proteomic paper parameters.

We have added a detailed description of the instrument settings and parameters to the methods section.

Reviewer 4:

This paper, which analyzes human dental calculus collected from archaeological specimens from the North Caucasus and nearby regions, represents an important and interesting set of data that sheds some light on the consumption of dairy products from domesticated herd animals. The research is aimed at answering an important question in the archaeology of the region and the results of the work will be of interest to the field. However, I cannot recommend the manuscript for publication in its current form.

Comment 18: The primary issue with the article, as currently written, is that it over-interprets the data that the authors have generated. The interpretations that they propose ignore some of the basic deficiencies, or more perhaps optimistically, challenges of their samples. The authors have tested dental calculus from 45 people across 29 sites from a time period spanning ~6,000 years. Needless to say, this is thin coverage – both temporally and spatially. The sample of 27 people from sites in the North Caucasus is closer to being sufficient, but the samples from the South Caucasus, Oka-Volga-Don, and eastern Urals are too small to support interpretations beyond random positive evidence of dairy consumption. Moreover, within the regional datasets, they have sampled only a small number of individuals from the same site (some from different phases). Thus, the dataset they've gathered cannot meaningfully address the possibility that there are dietary differences within communities.

The main focus of this study is on the North Caucasus region, and the goal was to analyze archaeological dental calculus proteins in order to identify direct positive evidence for milk protein consumption. Dental calculus from surrounding regions was analyzed to help situate the North Caucasus data within a broader context. While the samples analyzed from the South Caucasus, Oka-Volga-Don, and eastern Urals are far from comprehensive, they represent the first proteomic data generated for many of these sites, and we do not believe the paper would be strengthened by removing this data from the paper. Given their sample size, we have taken care to not over-interpret findings from these regions, and instead place the findings within their archaeological context so that others may build upon this work in future studies as more samples become available. We have updated our language when addressing the findings from these regions to note whether our data are consistent with other lines of archaeological evidence.

Comment 19: The authors have failed to directly state and support certain assumptions that appear to be at the foundations of their interpretations. It seems that they are working from the assumption that if there is dairying (be it consumption or a dairying “economy”), then everyone (or a very large percentage of people) would be consuming dairy products in such a way that dairy products would be detectable in the calculus. I am not sure this is a reasonable assumption, given that a recent paper on the method indicates that: “Little is currently known about how dietary proteins become trapped within dental calculus, and variation in this process may influence downstream protein recovery and identification success. Until we understand the degradation of these proteins, we cannot conclude that the absence of evidence is the evidence of absence.” (Hendy et al. 2018, 5).

We agree that small-scale, limited, or casual dairying may not be detectable in all members of a society. This is why we do not claim that an absence of observed milk peptides indicates a complete absence of dairying. Rather, we focus instead on where milk peptides are observed as positive evidence for dairying. Within the North Caucasus region, the dominant subsistence strategy on the steppe throughout the Bronze Age and Iron Age periods is known to have been pastoralism. This has been extensively documented through archaeological excavation and stable isotope investigations. What was not possible to conclusively demonstrate until now is whether this pastoralism included dairying. We show that the dental calculus of all tested Bronze Age and Iron Age individuals in the North Caucasus contained milk proteins. This is consistent with milk being a staple component of the pastoralist diet during these periods.

We are familiar with the quote that the reviewer has cited because a member of this study (CW) was one of the authors who wrote it. However, the reviewer has misunderstood the meaning of this statement. The point of the statement is to acknowledge that not all dietary proteins become incorporated into dental calculus. However, when dietary proteins are detected within dental calculus, it indicates that they were a component of the diet. Here we focus on locations and periods where we can confirm milk proteins were consumed.

Comment 20: Otherwise, how are we to understand what conclusions to draw when a single individual from their dataset does not show evidence of dairy consumption in their calculus? On one hand, if this individual is truly from a community that did not consume dairy, it is truly evidence of an absence of dairy consumption. But if we are dealing with a community where only 1 in 4 people consume dairy in a way where we would expect to be able to detect it in the dental calculus, then it isn't truly evidence of an absence of dairying. It is merely an artifact of choosing too small of sample to measure the prevalence of the event/trait in the population. This is true at the level of site and the level of the region and/or time period.

We believe there may be a misunderstanding here. The argument made by Hendy et al. 2018 is that paleoproteomics approaches are capable of positively identifying dietary components. However, negative data cannot be interpreted as straightforward evidence of absence. This is because dietary proteins may fail to incorporate into calculus for some individuals, and because some proteins may not sufficiently preserve for detection. In addition, not every individual builds up calculus. Aware of these limitations, we have structured this study to focus on positive evidence, for which paleoproteomics is a robust method. In the North Caucasus region, we detected milk protein in 26 of 27 individuals with calculus. More specifically, we detected it in 24 of 24 Bronze and Iron Age individuals and 2 of 3 Eneolithic individuals. From this, we conclude that livestock milk was produced and consumed by individuals spanning the Eneolithic, Bronze Age, and Iron Age periods. We are not sure why the reviewer is interpreting the absence of milk protein in one individual as conclusive evidence that that person did not consume milk. We do not make that argument in our study. Throughout our discussion, we have ensured that we have used cautious language when interpreting cases of milk protein detection at low frequency or negative recovery.

Comment 21: Likewise, the authors need to be clear about what level of consumption constitutes the phenomenon of interest. Does dairying only matter when it is everyone in the community? 75%? 50? 25%? Any consumption of milk whatsoever? The current experimental design is only adequate to one scenario – a very high-level of consistent dairy consumption. Connected to this issue is the slippage in how the authors use the term “dairying”, which admittedly, is a very common feature of the literature on prehistoric dairying. The authors should think carefully about what they mean when they say dairying. Is it merely any human consumption of nonhuman milk products? Or are they referring to a particular kind of pastoralist economy (with the attendant assumption that milk products would provide a greater portion of the diet or have a more important role in the pastoralist economy). The ability or inability of the presently-available data and methods to address either form of dairying needs to be explicitly addressed.

We agree with the reviewer that proteomics data alone could not be used to conclusively determine the subsistence economy of an ancient population. That is why in this study we incorporate our findings within a much broader and diverse range of archaeological evidence that together robustly support that the Bronze Age North Caucasus steppe populations practiced a mobile pastoralist subsistence economy. Moreover, we show that milk protein consumption did not begin in the Bronze Age, but rather traces its origins to previous Eneolithic populations in the region. By embedding our findings within this robust archaeological framework, we are able to provide evidence for how dairying began and developed through time in the region.

Prior work on the archaeology of the North Caucasus has already established that mobile pastoralism was the dominant form of Bronze Age subsistence economy on the steppe. This is well supported by archaeological, zooarchaeological, and isotopic evidence, and the published literature supporting this is discussed and cited in the text. Unlike populations south of the Caucasus or populations living in the mountains or mountain piedmont zone, there is no evidence for permanent settlements, agriculture, or agropastoralism in the North Caucasus steppe zone during the Bronze Age. Thus, Bronze Age populations in the steppe are considered to have practiced pastoralism but not agropastoralism, and they specifically practiced a form of mobile pastoralism focused on ruminant livestock. What was not known prior to this study was whether the form of mobile pastoralism practiced in the North Caucasus included or did not include dairying. Using our proteomics data, we have been able to show that these mobile pastoralists consumed foods containing livestock milk proteins, which could only have been produced through dairying (the production of livestock milk and its transformation into milk-based foods).

Putting this information together, we are able to say that previously published archaeological, zooarchaeological, and isotopic data robustly support that the Bronze Age North Caucasus populations living on the steppe were mobile pastoralists. Our proteomic evidence further clarifies their subsistence economy by showing that ruminant milk products were consumed, confirming that ruminant dairying was a component of their mobile pastoralism.

It is not currently possible to provide absolute quantifications of the volume of milk consumed by an individual from the number of milk PSMs identified in dental calculus. This is in part due to variation in dietary protein incorporation during life and protein preservation after death. For this reason, although we quantify the number of milk PSMs identified per individual to show the strength and reliability of the measurements, we interpret the results collectively by region and time period to confirm the presence of specific livestock milk in the diet.

It is currently not possible to precisely define the exact fraction of the community that consumed milk products, given the challenges in interpreting negative data. However, in the North Caucasus region, we detected milk protein in 26 of 27 individuals with calculus, which indicates that it was a major dietary component for these communities. Of all of the ancient and modern populations studied to date, the Bronze Age North Caucasus populations have the highest prevalence (proportion of milk-positive individuals among tested individuals) and highest abundance (absolute number of milk PSMs) yet identified. This, combined with their known mobile pastoralist lifestyle, suggests that the North Caucasus Bronze Age societies practiced a form of mobile dairy pastoralism.

Comment 22: Lines 277-278: The authors write that “Surprisingly, however, the dairy economy remained exclusively focused on sheep.” This claim is made in reference to the data the authors have collected from 7 individuals across 1,000 years. This is an overinterpretation, since the dataset can only positively identify presence, especially without any detailed zooarchaeological data from these sites to support this conclusion.

We have revised this sentence to state: “Surprisingly, however, the dairy economy retained an apparent focus on sheep.”

Comment 23: Lines 298-299: The authors state that “Most individuals of the Middle Bronze Age Catacomb, NCC, Late NCC, and Lola cultures consumed the dairy products of two or three livestock species.” As written, in the larger context of the article, this statement is misleading (and is likely to be misunderstood). More adequate would be something like: “Most of the individuals from the Middle Bronze Age Catacomb, NCC, Late NCC, and Lola cultures that were included in the study [alternate version: that were tested] consumed dairy products from two or three livestock species.”

We have updated this sentence as the reviewer suggested to state: “Most of the individuals from the Middle Bronze Age Catacomb, NCC, Late NCC, and Lola cultures tested in this study consumed dairy products from two or three livestock species.”

Comment 24: Lines 311-313: The authors write ““Our results suggest that during the Lola period, water-demanding cattle began to decrease in economic importance from the preceding Catacomb and NCC periods, as

only one out of six Lola individuals yielded evidence for cattle milk consumption.” This claim is not supported by the evidence. Cattle could have been used for non-milk purposes, which is precisely what the authors suggested previously in the paper when they interpreted the Maykop data. Without any corroboration from zooarchaeological data, the climate explanation cannot be given precedence.

We have revised this sentence to state: “Our results suggest that during the Lola period, water-demanding cattle may have decreased in dairying importance from the preceding Catacomb and NCC periods, as only one out of six Lola individuals yielded evidence for cattle milk consumption.”

Comment 25: Lines 377-380: The claims the authors make here are not tenable because there cannot be a legitimate comparison between the datasets they present from the North and South Caucasus. The data from the North and South Caucasus are divergent in terms of their sample sizes and in terms of the adequacy of regional coverage.

We have revised this statement to remove the claim.

Comment 26: Line 388: The authors write about “the near-complete lack of evidence for ruminant milk consumption from the seven individuals” that they tested from the Oka-Volga-Don region. This line could be re-written, and would perhaps be more accurate, as “14% of individuals tested showed evidence of ruminant milk consumption.”

We have revised our phrasing to make our argument clearer. Among the Oka-Volga-Don individuals in our study, the only individual who tested positive for milk protein was a Catacomb-associated individual with cultural links to the steppe zone. Thus, it is likely that dairying was introduced through the arrival of Catacomb groups from the steppe zone of the North Caucasus who practiced dairy pastoralism, or by contact with Catacomb groups who culturally spread their knowledge. The remaining sites do not show archaeological or zooarchaeological evidence of mobile pastoralism.

Comment 27: Lines 409-411: The authors write “Dairying was integral for the spread of animal husbandry by groups crossing the Caucasus mountains from south to north during the Eneolithic”. The evidence presented in the paper is not able to support this claim. Not only is the data from the South Caucasus highly partial, it is also basically contemporaneous with the data they’ve collected for the North.

We agree that proteomic evidence alone would not be able to support this claim. However, here we are incorporating our proteomic data with a body of previously published archaeological and paleogenomic data that have already established direct genetic links between Chalcolithic populations south of the Caucasus mountains and Eneolithic populations living in the piedmont zone on the north side of the Caucasus mountains. Thus, our protein data clarifies and refines previous evidence in the region.

Previous research in the region has shown that new groups appear in the piedmont zone of the North Caucasus during the Eneolithic, and that these new groups have an ancestry linking them to South Caucasus populations and that they have shared forms of food production with South Caucasus groups, including specific domesticates that link them to the south. Thus, archaeological, zooarchaeological, and paleogenomic evidence points to the movement of groups across the Caucasus mountains, and that these groups brought with them a specific package of domesticates that included sheep, goat, and cattle livestock. Our proteomic data confirm that these earliest North Caucasus Eneolithic groups consumed dairy products - with 2 of 3 testing positive for milk proteins. Regardless of how much milk protein they consumed, dairying was at least known to them and practiced by them. We also show that dairying was practiced by at least some contemporaneous groups south of the Caucasus, confirming that dairying (production of human food from animal milk) was a known technology in the South Caucasus. Clearly, the number of South Caucasus sites and individuals we were able to analyze in this study is underpowered to determine the full extent (temporal and geographic) of dairying in the South Caucasus, but it does represent the very first proteomic data for the region and it establishes that dairying was known and practiced by at least the Chalcolithic period at sites like Alkhantepe. We hope that future studies (that have not been burdened with the extreme logistical challenges of the global COVID-19 pandemic) will be able to greatly expand proteomic research in this important region.

Due to the limited number of dental calculus samples available for the South Caucasus, we have constrained our interpretations of the data to those that we can robustly support using previously published archaeological and paleogenomic data together with our new proteomic data.

Comment 28: Lines 413-414: The authors write “Initial pastoralist strategies were specialized, with dairying focused on sheep and cattle used for traction”. This overstates the evidence. The claim might be applicable if limited to the Maykop, though that is also probably still an overreach.

We have updated this sentence to state: “Initial pastoralist strategies focused on sheep dairying and cattle traction, while fully mobile pastoralism arose for the first time during the Yamnaya period.”

Comment 29: Figure 3: This figure is very misleading and probably should be abandoned. It does not show what is missing from the datasets (in terms of temporal and regional coverage), producing an incorrect impression. Particularly troubling are: 1) the lack of any data from Georgia or Armenia, 2) the absence of any data from anywhere in the South Caucasus between 3500-2500 BC, 3) the absence of data from the North Caucasus prior to ~ 4500 BC, and 4) the fact that there is only data for the E. Urals for one period. Overall, the figure presents a misleading picture of the available data and is highly likely to be misinterpreted. It might be possible to make a version of this figure that only focuses on the North Caucasus, since that region has more adequate coverage.

We strongly disagree with this opinion. Figure 3 is a summary of our data through time. All individuals analyzed in this study are shown on the maps (together with whether they yielded milk proteins or not), and thus we feel this is a transparent representation of both our results and the extent of our geographic and temporal sampling. We appreciate that the reviewer is concerned that particular sites and regions were not represented in our study (particularly in the South Caucasus), and we specifically designed this figure to make these gaps visible so that readers can clearly see what was and was not sampled in the current study, so future studies can address these

geographic and temporal gaps. To make the value of Figure 3 clearer, we have updated the time periods shown in each panel to better reflect the changing patterns of milk consumption through time.

Comment 30: Finally, my reading of the manuscript has also left me with some more specific methodological questions that ideally would be addressed somewhere (possibly in supplementary materials, since word counts are often quite limiting).

- 1) Is all of the available calculus sampled? Or is there a distinct procedure for sampling a portion of the available calculus?
- 2) Are the authors able to suggest how much time is averaged in the sample calculus? What factors might influence inter-individual variation in the amount of time represented by the sampled calculus?
- 3) If the calculus sampled represents a somewhat shorter process of accumulation and averaging, how might any seasonality in milk consumption or the possibility that people of different ages consumed more or less dairy products impact the interpretation of data produced by this method?

For each individual, we sampled and analyzed 5-10 mg of calculus. This is approximately how much is typically found on 1-2 adult teeth. We preferentially analyzed all calculus present on a single tooth, and if this was not sufficient to reach 5mg, we sampled a second tooth. For adults, the diet represented is that consumed after the loss of the deciduous dentition and the eruption of the permanent dentition until death. For juveniles it represents the diet consumed prior to death. Calculus accumulation is typically greater after puberty than before puberty. While calculus does form incrementally, annual accumulations are far below that which was analyzed in this study, and thus the amount of calculus we sampled represents the cumulative diet over many years, and for most individuals over decades. There is no seasonal effect, as the amount analyzed represents many years of accumulation. Inter-individual variation is expected to relate only to the age of the individual. A person who died at age 30 is expected to have 10-15 years of calculus growth, whereas an individual who died at age 60 is expected to have 40-45 years of calculus growth. In this way, the overall period of dietary representation is relatively long, as is also true for bone collagen stable isotope studies, except that dental calculus does not remodel and thus represents the cumulative adult diet rather than only the past 12-15 years, which is the average turnover time of bone collagen.

Decision Letter, first revision:

20th December 2021

Dear Christina,

Thank you for submitting your revised manuscript "Emergence and intensification of dairying in the Caucasus and Eurasian steppes" (NATECOLEVOL-210914661A). It has now been seen again by the

original reviewers and their comments are below. The reviewers find that the paper has improved in revision, and therefore we'll be happy in principle to publish it in Nature Ecology & Evolution, pending minor revisions to satisfy the reviewers' final requests and to comply with our editorial and formatting guidelines. Regarding the former, I think if you can address the two issues reviewer 4 finds to be sticking points (using more nuanced language in terms of generalising to wider societies from the number of samples obtained and overhauling figure 3) we should be good to go.

We are now performing detailed checks on your paper and will send you a checklist detailing our editorial and formatting requirements in about a week (note--this is our usual target but we are now working with skeleton staff over the next couple of weeks so it may actually be January before we can get this to you). Please do not upload the final materials and make any revisions until you receive this additional information from us.

[REDACTED]

Reviewer #1 (Remarks to the Author):

The authors solved most concerns. However, a few points need more thinking, specially linking with Wilkin's paper.

1) During the Eneolithic period, the authors' dairy finding mainly focused on the North Caucasus piedmont zone, while the Wilkin's paper factually focused on the Oka-Volga-Don zone with rare dairy findings. I think there is no conflict, maybe dairying just didn't spread to the further north area during this period.

2) In the early Yamnaya culture, the authors' dairying finding is consistent with Wilkin's paper. Both studies found the sheep is the main source of dairying, although the research regions are different. But both work together will promote understanding the role of sheep dairying in the rise of mobile pastoralism.

3) There is a climate event around 5300 BP, when Yamnaya culture or mobile pastoralism rised. The authors should pay more attention to discuss about its impact.

4) The milk consumption is diversified between Early Bronze Age and Middle Bronze Age. Although it's difficult to find the settlement evidence about social change. However, the mortuary contexts including tomb size, burial objects, could also provide some important information about social structure or complexity. I think the authors should discuss more with archaeologists about the social change during the transition period.

Reviewer #2 (Remarks to the Author):

The authors have very thoroughly and thoughtfully addressed all the points raised by the reviewers and made appropriate adjustments to the text. In the original version, whilst generally excellent, there

was a degree of overreach in a few conclusions. I am now very content that there is additional specificity regarding the conclusions and appropriate clarifications have been added. This is an important study that adds very significant detail to what is currently known. I believe it will be much cited and now recommend publication.

Reviewer #3 (Remarks to the Author):

I recommend publication of your manuscript. I still find that your results will provide valuable and novel details within the specific research area and I do not question the presented data or your conclusions.

I do have comments, that I suggest you look at.

As you write yourself in the rebuttal 'we have focused our data analysis and interpretation on dietary proteins present within the dental calculus proteome', your analysis is done only on part of the proteome. You do not 'investigate the dental calculus proteomes' (P2 L70), but dietary proteins. On the other hand, you did apparently look into other proteins in the dataset, since you still state 'All dental calculus samples yielded proteomes consistent with an oral microbiome profile' (P4 L184). Maybe this absolutely true, but it is not apparent from only the milk proteins that you present in the paper.

Regarding comment 11 I believe it is very dataset dependent, and I agree that it can go both ways. <https://www.ncbi.nlm.nih.gov/pmc/articles/PMC4748730/>

Rosa Rakownikow Jersie-Christensen

Reviewer #4 (Remarks to the Author):

I want to begin by thanking the authors for their careful attention to and detailed response to my comments on the earlier version of the article. I want to emphasize, however, that they misunderstood many of the comments (#18-21). In focusing on the number of individuals sampled at individual sites and across regions and time periods, I was not questioning the validity or ability of the proteomic method to successfully detect the consumption of dairy products. I was questioning the adequacy and representativeness of the sampling of individuals from archaeological sites and regions. The methods used in this paper are powerful and important, but that does not mean they are not impacted by the same basic issues of sampling that all archaeological analyses are.

I am pleased to see that the authors have followed my suggestions for moderating many of the over-extended claims that I highlighted in the previous draft. But I am disappointed that they did not engage seriously with my request that they separate out claims about dairy consumption and a dairying economy. I think that would make this paper a much stronger contribution to the literature – setting the bar for future efforts.

For instance, in lines 163-4, the authors write "We find that dairy products were consumed in the Caucasus from the late 5th millennium BCE onwards and that a dairy-based subsistence characterizes even the Eneolithic populations in the piedmont and steppe zones." This study has positively identified milk proteins in 3 individuals from Eneolithic sites in the North Caucasus. This choice of phrasing, absent a detailed discussion of the differences (both sociological and evidentiary) between any dairy

consumption and the suggestion that dairy is a primary component of subsistence, is misleading.

Similarly, the authors use the term "dairying economy" rather imprecisely. Mere evidence of dairy consumption is not the same thing as economic practices that rely heavily on dairy production. It is important for the study of pastoralism, particularly mobile pastoralism, to mind the difference between the two – especially if we are to determine the role that different levels of dairy consumption and different kinds of pastoralist economies played in the prehistory of mobile pastoralists in Eurasia. If the authors are unwilling or unable to add such a discussion, I would strongly recommend avoiding the phrase "dairy-based subsistence" and "dairying economy" altogether, and suggest using the more accurate "dairy consumption" in its place.

Additionally, the authors have introduced new problems to the paper. Lines 386-9 repeat the problem I outlined in the earlier draft. It is not a valid comparison, because the level of regional coverage between samples from the South and North Caucasus is wildly divergent. Put simply, it is comparing apples and oranges, and is not in any way a meaningful comparison.

I am also disappointed at the lack of consideration given to the criticisms of Figure 3. It doesn't add meaningfully to the information provided in Figure 1 and gives a very misleading impression to the reader. While I think it should be scrapped entirely, I will offer two additional suggestions for improving it: 1) limiting the figure to the North Caucasus (i.e. zooming in on the base map), to avoid giving erroneous impressions about the situation in Georgia and Azerbaijan or 2) adding in dark dots for sites (or even just major sites) for each time/region that are not included in the samples tested in this paper.

To summarize, I think the most important things that remain to be addressed are:

- 1) The language of "dairy-based subsistence" and "dairy economy"
- 2) Figure 3

Our ref: NATECOLEVOL-210914661A

21st January 2022

Dear Dr. Warinner,

Thank you for your patience as we've prepared the guidelines for final submission of your Nature Ecology & Evolution manuscript, "Emergence and intensification of dairying in the Caucasus and Eurasian steppes" (NATECOLEVOL-210914661A). Please carefully follow the step-by-step instructions provided in the attached file, and add a response in each row of the table to indicate the changes that you have made. Please also check and comment on any additional marked-up edits we have proposed within the text. Ensuring that each point is addressed will help to ensure that your revised manuscript can be swiftly handed over to our production team.

****We would like to start working on your revised paper, with all of the requested files and forms, as soon as possible (preferably within two weeks). Please get in contact with us immediately if you anticipate it taking more than two weeks to submit these revised files.****

In recognition of the time and expertise our reviewers provide to Nature Ecology & Evolution's editorial process, we would like to formally acknowledge their contribution to the external peer review of your manuscript entitled "Emergence and intensification of dairying in the Caucasus and Eurasian steppes". For those reviewers who give their assent, we will be publishing their names alongside the published article.

Nature Ecology & Evolution offers a Transparent Peer Review option for new original research manuscripts submitted after December 1st, 2019. As part of this initiative, we encourage our authors to support increased transparency into the peer review process by agreeing to have the reviewer comments, author rebuttal letters, and editorial decision letters published as a Supplementary item. When you submit your final files please clearly state in your cover letter whether or not you would like to participate in this initiative. Please note that failure to state your preference will result in delays in accepting your manuscript for publication.

Cover suggestions

As you prepare your final files we encourage you to consider whether you have any images or illustrations that may be appropriate for use on the cover of Nature Ecology & Evolution.

Please submit your suggestions, clearly labeled, along with your final files. We'll be in touch if more

information is needed.

Nature Ecology & Evolution has now transitioned to a unified Rights Collection system which will allow our Author Services team to quickly and easily collect the rights and permissions required to publish your work. Approximately 10 days after your paper is formally accepted, you will receive an email in providing you with a link to complete the grant of rights. If your paper is eligible for Open Access, our Author Services team will also be in touch regarding any additional information that may be required to arrange payment for your article.

Please note that *Nature Ecology & Evolution* is a Transformative Journal (TJ). Authors may publish their research with us through the traditional subscription access route or make their paper immediately open access through payment of an article-processing charge (APC). Authors will not be required to make a final decision about access to their article until it has been accepted. [Find out more about Transformative Journals](https://www.springernature.com/gp/open-research/transformative-journals)

Authors may need to take specific actions to achieve compliance with funder and institutional open access mandates. For submissions from January 2021, if your research is supported by a funder that requires immediate open access (e.g. according to [Plan S principles](https://www.springernature.com/gp/open-research/plan-s-compliance)) then you should select the gold OA route, and we will direct you to the compliant route where possible. For authors selecting the subscription publication route our standard licensing terms will need to be accepted, including our [self-archiving policies](https://www.springernature.com/gp/open-research/policies/journal-policies). Those standard licensing terms will supersede any other terms that the author or any third party may assert apply to any version of the manuscript.

[REDACTED]

[REDACTED]

Reviewer #1:

Remarks to the Author:

The authors solved most concerns. However, a few points need more thinking, specially linking with

Wilkin's paper.

1) During the Eneolithic period, the authors' dairy finding mainly focused on the North Caucasus piedmont zone, while the Wilkin's paper factually focused on the Oka-Volga-Don zone with rare dairy findings. I think there is no conflict, maybe dairying just didn't spread to the further north area during this period.

2) In the early Yamnaya culture, the authors' dairying finding is consistent with Wilkin's paper. Both studies found the sheep is the main source of dairying, although the research regions are different. But both work together will promote understanding the role of sheep dairying in the rise of mobile pastoralism.

3) There is a climate event around 5300 BP, when Yamnaya culture or mobile pastoralism rised. The authors should pay more attention to discuss about its impact.

4) The milk consumption is diversified between Early Bronze Age and Middle Bronze Age. Although it's difficult to find the settlement evidence about social change. However, the mortuary contexts including tomb size, burial objects, could also provide some important information about social structure or complexity. I think the authors should discuss more with archaeologists about the social change during the transition period.

Reviewer #2:

Remarks to the Author:

The authors have very thoroughly and thoughtfully addressed all the points raised by the reviewers and made appropriate adjustments to the text. In the original version, whilst generally excellent, there was a degree of overreach in a few conclusions. I am now very content that there is additional specificity regarding the conclusions and appropriate clarifications have been added. This is an important study that adds very significant detail to what is currently known. I believe it will be much cited and now recommend publication.

Reviewer #3:

Remarks to the Author:

I recommend publication of your manuscript. I still find that your results will provide valuable and novel details within the specific research area and I do not question the presented data or your conclusions.

I do have comments, that I suggest you look at.

As you write yourself in the rebuttal 'we have focused our data analysis and interpretation on dietary proteins present within the dental calculus proteome', your analysis is done only on part of the proteome. You do not 'investigate the dental calculus proteomes' (P2 L70), but dietary proteins.

On the other hand, you did apparently look into other proteins in the dataset, since you still state 'All dental calculus samples yielded proteomes consistent with an oral microbiome profile' (P4 L184). Maybe this absolutely true, but it is not apparent from only the milk proteins that you present in the paper.

Regarding comment 11 I believe it is very dataset dependent, and I agree that it can go both ways. <https://www.ncbi.nlm.nih.gov/pmc/articles/PMC4748730/>

Rosa Rakownikow Jersie-Christensen

Reviewer #4:

Remarks to the Author:

I want to begin by thanking the authors for their careful attention to and detailed response to my comments on the earlier version of the article. I want to emphasize, however, that they misunderstood many of the comments (#18-21). In focusing on the number of individuals sampled at individual sites and across regions and time periods, I was not questioning the validity or ability of the proteomic method to successfully detect the consumption of dairy products. I was questioning the adequacy and representativeness of the sampling of individuals from archaeological sites and regions. The methods used in this paper are powerful and important, but that does not mean they are not impacted by the same basic issues of sampling that all archaeological analyses are.

I am pleased to see that the authors have followed my suggestions for moderating many of the over-extended claims that I highlighted in the previous draft. But I am disappointed that they did not engage seriously with my request that they separate out claims about dairy consumption and a dairying economy. I think that would make this paper a much stronger contribution to the literature – setting the bar for future efforts.

For instance, in lines 163-4, the authors write “We find that dairy products were consumed in the Caucasus from the late 5th millennium BCE onwards and that a dairy-based subsistence characterizes even the Eneolithic populations in the piedmont and steppe zones.” This study has positively identified milk proteins in 3 individuals from Eneolithic sites in the North Caucasus. This choice of phrasing, absent a detailed discussion of the differences (both sociological and evidentiary) between any dairy consumption and the suggestion that dairy is a primary component of subsistence, is misleading.

Similarly, the authors use the term “dairying economy” rather imprecisely. Mere evidence of dairy consumption is not the same thing as economic practices that rely heavily on dairy production. It is important for the study of pastoralism, particularly mobile pastoralism, to mind the difference between the two – especially if we are to determine the role that different levels of dairy consumption and different kinds of pastoralist economies played in the prehistory of mobile pastoralists in Eurasia. If the authors are unwilling or unable to add such a discussion, I would strongly recommend avoiding the phrase “dairy-based subsistence” and “dairying economy” altogether, and suggest using the more accurate “dairy consumption” in its place.

Additionally, the authors have introduced new problems to the paper. Lines 386-9 repeat the problem I outlined in the earlier draft. It is not a valid comparison, because the level of regional coverage between samples from the South and North Caucasus is wildly divergent. Put simply, it is comparing apples and oranges, and is not in any way a meaningful comparison.

I am also disappointed at the lack of consideration given to the criticisms of Figure 3. It doesn't add meaningfully to the information provided in Figure 1 and gives a very misleading impression to the reader. While I think it should be scrapped entirely, I will offer two additional suggestions for improving it: 1) limiting the figure to the North Caucasus (i.e. zooming in on the base map), to avoid giving erroneous impressions about the situation in Georgia and Azerbaijan or 2) adding in dark dots for sites (or even just major sites) for each time/region that are not included in the samples tested in this paper.

To summarize, I think the most important things that remain to be addressed are:

- 1) The language of “dairy-based subsistence” and “dairy economy”
- 2) Figure 3

Author Rebuttal, first revision:

**Point-by-Point Response to Reviewer Comments for NATECOLEVOL-210914661
“Emergence and intensification of dairying in the Caucasus and Eurasian steppes”**

We received 11 specific comments from four reviewers. We present our responses to each comment below. Changes made are marked in blue in the revised manuscript file.

Reviewer 1:

The authors solved most concerns. However, a few points need more thinking, specially linking with Wilkin's paper.

Comment 1: During the Eneolithic period, the authors' dairy finding mainly focused on the North Caucasus piedmont zone, while the Wilkin's paper factually focused on the Oka-Volga-Don zone with rare dairy findings. I think there is no conflict, maybe dairying just didn't spread to the further north area during this period.

We agree that this assessment is consistent with the data.

Comment 2: In the early Yamnaya culture, the authors' dairying finding is consistent with Wilkin's paper. Both studies found the sheep is the main source of dairying, although the research regions are different. But both work together will promote understanding the role of sheep dairying in the rise of mobile pastoralism.

We agree that both studies indicate an early reliance on sheep milk. We cite the findings of the Wilkin et al. study in our discussion, but we are unable to integrate the study's data into our results section because data reporting discrepancies and missing files prevent us from reanalyzing the data. We informed the corresponding authors and the handling editor of Wilkin et al. 2021, but the situation could not be clarified by the time of this resubmission.

Comment 3: There is a climate event around 5300 BP, when Yamnaya culture or mobile pastoralism rised. The authors should pay more attention to discuss about its impact.

Current evidence for a climate shift at ca. 3300 BCE (ca. 5300 BP, coinciding with initial Yamnaya expansions) is minor and not easily distinguishable from other minor climatic variation throughout the period. Overall, pollen data from the steppe zone indicate dry steppe vegetation and a continental climate from ca. 4000-2500 BCE. Instability, which indicates a short episode of aridification (i.e., less precipitation, lower temperatures), is recorded in pollen data between the lower Dona and Volga between ca. 3500-3400 BCE (Kremenetsky 1997; Shishlina 2008; Richards et al. 2014). This seems, however, to reflect local rather than regional events. There is evidence in Southeast Europe, Anatolia and the North Caucasus for fluctuations in precipitation and temperature between 4000-2000 BCE, with a slightly negative deflection for precipitation and temperatures ca. 3300 BCE (Davis et al. 2003). The steppe environmental data, however, does not record this as a lasting negative shift (see Shishlina 2008, 220). The vegetation of the Caucasus mountains fluctuates much less and indicates few overall changes (Conor and Kvavadze 2009). In the Lesser Caucasus, drier conditions and an expansion of open grasslands started ca. 3700 BCE but only become established in the 3rd mill. BCE (e.g. Leroyer et al. 2015).

Overall, current evidence for climate change being an influential factor in initial Yamnaya expansions is ambiguous and a strong emphasis would seem overly deterministic. Instead, evidence for climate change in the North Caucasus is clearer during the 3rd mill. BCE. During this period, we observe a shift toward dairy herd diversification among late Yamnaya, NCC, and Catacomb groups. This in initial diversification also overlaps in time with subsequent Yamnaya expansions into southeastern Europe as well as the parallel rise and expansion of the Corded Ware complex across northeastern and central Europe. We have added additional citations regarding climate patterns during the 4th and 3rd millennia BCE, and we have expanded the discussion of the Yamnaya to include the Corded Ware.

References:

- Kremenetsky, K.V., 1997. Prirodnaya Obstanovka Golosena Na Nizhnem Donu i v Kalmykii. Step' i Kavkaz. Trudy Geologicheskogo Instiuta, Moscow 97, pp. 30–45 (in Russian)

- Shishlina, N., 2008. *Reconstruction of the Bronze Age of the Caspian steppes: Life styles and life ways of pastoral nomads*. Archaeopress.
- Richards, K., Bolikhovskaya, N.S., Hoogendoorn, R.M., Kroonenberg, S.B., Leroy, S.A. and Athersuch, J., 2014. Reconstructions of deltaic environments from Holocene palynological records in the Volga delta, northern Caspian Sea. *The Holocene*, 24(10), pp.1226-1252.
- Davis, B.A., Brewer, S., Stevenson, A.C. and Guiot, J., 2003. The temperature of Europe during the Holocene reconstructed from pollen data. *Quaternary science reviews*, 22(15-17), pp.1701-1716.
- Connor, S.E. and Kvavadze, E.V., 2009. Modelling late Quaternary changes in plant distribution, vegetation and climate using pollen data from Georgia, Caucasus. *Journal of Biogeography*, 36(3), pp.529-545.
- Leroyer, C., Joannin, S., Aoustin, D., Ali, A.A., Peyron, O., Ollivier, V., Tozalakyan, P., Karakhanyan, A. and Jude, F., 2016. Mid Holocene vegetation reconstruction from Vanevan peat (south-eastern shore of Lake Sevan, Armenia). *Quaternary International*, 395, pp.5-18.

Comment 4: The milk consumption is diversified between Early Bronze Age and Middle Bronze Age. Although it's difficult to find the settlement evidence about social change. However, the mortuary contexts including tomb size, burial objects, could also provide some important information about social structure or complexity. I think the authors should discuss more with archaeologists about the social change during the transition period.

Within the North Caucasus, an archaeological transition is clear in the early 3rd millennium BCE when Maykop, Steppe Maykop, and Early Yamnaya groups declined and were replaced by North Caucasus Culture, Catacomb, and Late Yamnaya groups. Although settlement data are not available for this critical period because all North Caucasus groups at that time were mobile, these groups differed in the environments they exploited and in their mortuary treatment of the dead. In general, the Middle Bronze Age groups were more mobile and exploited larger pastures than Early Bronze Age groups. The NCC groups expanded into the mountains, while the Late Yamnaya and Catacomb groups expanded into more remote areas of the steppe. Overall, burial richness decreased from the Early to Middle Bronze Age: the NCC groups had fewer and less diverse grave goods than prior Maykop groups in the mountain piedmont area, but the differences between the Early Bronze Age (Steppe Maykop and Early Yamnaya) and Middle Bronze Age (Catacomb and Late Yamnaya) groups in the steppe were less pronounced. Some new features appeared during the Middle Bronze Age, such as the appearance of burial mound lines and a general increase in the number of burials.

Overall, one of the important results of our study is that we were able to detect a clear shift in dairying practices that otherwise did not leave a strong archaeological footprint due to the nature of surviving archaeological sites in the region (which are almost entirely restricted to mortuary contexts). Presumably, this dairying shift was linked to other social changes, but they are archaeologically invisible or difficult to discern. We have discussed this issue at length among the archaeologists on the study, and they agree that the clearest surviving archaeological correlate with dairy diversification in the North Caucasus is a general increase in mobility. The period of dairy diversification also overlaps in time with the emergence of the Corded Ware cultural complex in Europe, and the two events may be related to broader social or economic changes occurring within steppe and forest steppe pastoralist societies at the time. We have updated the Discussion to make these connections clearer, and we have provided detailed information about the archaeological context, tomb contents, and other relevant features for each burial in this study in the supplementary text. However, the archaeologist members of our study are reluctant to further speculate on possible changes to social structure or complexity given the limited nature of current archaeological evidence.

Reviewer 2:

Comment 5: The authors have very thoroughly and thoughtfully addressed all the points raised by the reviewers and made appropriate adjustments to the text. In the original version, whilst generally excellent, there was a degree of overreach in a few conclusions. I am now very content that there is additional specificity regarding the conclusions and appropriate clarifications have been added. This is an important study that adds very significant detail to what is currently known. I believe it will be much cited and now recommend publication.

We thank the reviewer for their positive assessment.

Reviewer 3:

Comment 6: I recommend publication of your manuscript. I still find that your results will provide valuable and novel details within the specific research area and I do not question the presented data or your conclusions. I do have comments, that I suggest you look at. As you write yourself in the rebuttal 'we have focused our data analysis and interpretation on dietary proteins present within the dental calculus proteome', your analysis is done only on part of the proteome. You do not 'investigate the dental calculus proteomes' (P2 L70), but dietary proteins.

We have updated the sentences in which this phrase appears to state: "...we analyzed dietary proteins within the dental calculus proteomes..."

Comment 7: On the other hand, you did apparently look into other proteins in the dataset, since you still state 'All dental calculus samples yielded proteomes consistent with an oral microbiome profile' (P4 L184). Maybe this absolutely true, but it is not apparent from only the milk proteins that you present in the paper. Regarding comment 11 I believe it is very dataset dependent, and I agree that it can go both ways. <https://www.ncbi.nlm.nih.gov/pmc/articles/PMC4748730/>

We agree that it is database dependent, and we have provided the raw data so that others may further investigate the remaining proteome if they wish to do so.

Reviewer 4:

I want to begin by thanking the authors for their careful attention to and detailed response to my comments on the earlier version of the article. I want to emphasize, however, that they misunderstood many of the comments (#18-21). In focusing on the number of individuals sampled at individual sites and across regions and time periods, I was not questioning the validity or ability of the proteomic method to successfully detect the consumption of dairy products. I was questioning the adequacy and representativeness of the sampling of individuals from archaeological sites and regions. The methods used in this paper are powerful and important, but that does not mean they are not impacted by the same basic issues of sampling that all archaeological analyses are. I am pleased to see that the authors have followed my suggestions for moderating many of the over-extended claims that I highlighted in the previous draft. But I am disappointed that they did not engage seriously with my request that they separate out claims about dairy consumption and a dairying economy. I think that would make this paper a much stronger contribution to the literature – setting the bar for future efforts.

For instance, in lines 163-4, the authors write "We find that dairy products were consumed in the Caucasus from the late 5th millennium BCE onwards and that a dairy-based subsistence characterizes even the Eneolithic populations in the piedmont and steppe zones." This study has positively identified milk proteins in 3 individuals from Eneolithic sites in the North Caucasus. This choice of phrasing, absent a detailed discussion of the differences (both sociological and evidentiary) between any dairy consumption and the suggestion that dairy is a primary component of subsistence, is misleading.

We have updated this sentence to state: "We find that dairy products were consumed in the Caucasus from the late 5th millennium BCE onwards and that a dairy-inclusive subsistence characterizes even the Eneolithic populations in the piedmont and steppe zones."

Comment 8: Similarly, the authors use the term "dairying economy" rather imprecisely. Mere evidence of dairy consumption is not the same thing as economic practices that rely heavily on dairy production. It is important for the study of pastoralism, particularly mobile pastoralism, to mind the difference between the two – especially if we are to determine the role that different levels of dairy consumption and different kinds of pastoralist economies played in the prehistory of mobile pastoralists in Eurasia. If the authors are unwilling or unable to add such a discussion, I would strongly recommend avoiding the phrase "dairy-based subsistence" and "dairying economy" altogether, and suggest using the more accurate "dairy consumption" in its place.

We have removed the terms “dairy-based subsistence” and “dairying economy.”

Comment 9: Additionally, the authors have introduced new problems to the paper. Lines 386-9 repeat the problem I outlined in the earlier draft. It is not a valid comparison, because the level of regional coverage between samples from the South and North Caucasus is wildly divergent. Put simply, it is comparing apples and oranges, and is not in any way a meaningful comparison.

We have removed this sentence.

Comment 10: I am also disappointed at the lack of consideration given to the criticisms of Figure 3. It doesn't add meaningfully to the information provided in Figure 1 and gives a very misleading impression to the reader. While I think it should be scrapped entirely, I will offer two additional suggestions for improving it: 1) limiting the figure to the North Caucasus (i.e. zooming in on the base map), to avoid giving erroneous impressions about the situation in Georgia and Azerbaijan or 2) adding in dark dots for sites (or even just major sites) for each time/region that are not included in the samples tested in this paper.

We believe Figure 3 adds value to the study by making the spatial and temporal relationships clearer. We have now limited Figure 3 to the North Caucasus. Option 2 strikes us as being infeasible and thus an unreasonable request.

Comment 11: To summarize, I think the most important things that remain to be addressed are:
1) The language of “dairy-based subsistence” and “dairy economy”
2) Figure 3

We have removed the terms “dairy-based subsistence” and “dairy economy”. We have updated Figure 3 to focus only on the North Caucasus.

Final Decision Letter:

7th February 2022

Dear Christina,

We are pleased to inform you that your Article entitled "Emergence and intensification of dairying in the Caucasus and Eurasian steppes", has now been accepted for publication in Nature Ecology & Evolution.

Over the next few weeks, your paper will be copyedited to ensure that it conforms to Nature Ecology and Evolution style. Once your paper is typeset, you will receive an email with a link to choose the appropriate publishing options for your paper and our Author Services team will be in touch regarding any additional information that may be required

You will not receive your proofs until the publishing agreement has been received through our system

Due to the importance of these deadlines, we ask you please us know now whether you will be difficult to contact over the next month. If this is the case, we ask you provide us with the contact information (email, phone and fax) of someone who will be able to check the proofs on your behalf, and who will be available to address any last-minute problems . Once your paper has been scheduled for online publication, the Nature press office will be in touch to confirm the details.

Acceptance of your manuscript is conditional on all authors' agreement with our publication policies (see www.nature.com/authors/policies/index.html). In particular your manuscript must not be published elsewhere and there must be no announcement of the work to any media outlet until the publication date (the day on which it is uploaded onto our web site).

Please note that *Nature Ecology & Evolution* is a Transformative Journal (TJ). Authors may publish their research with us through the traditional subscription access route or make their paper immediately open access through payment of an article-processing charge (APC). Authors will not be required to make a final decision about access to their article until it has been accepted. [Find out more about Transformative Journals](https://www.springernature.com/gp/open-research/transformative-journals)

Authors may need to take specific actions to achieve [compliance](https://www.springernature.com/gp/open-research/funding/policy-compliance-faqs) with funder and institutional open access mandates. For submissions from January 2021, if your research is supported by a funder that requires immediate open access (e.g. according to [Plan S principles](https://www.springernature.com/gp/open-research/plan-s-compliance)) then you should select the gold OA route, and we will direct you to the compliant route where possible. For authors selecting the subscription publication route our standard licensing

terms will need to be accepted, including our [self-archiving policies](https://www.springernature.com/gp/open-research/policies/journal-policies). Those standard licensing terms will supersede any other terms that the author or any third party may assert apply to any version of the manuscript.

We welcome the submission of potential cover material (including a short caption of around 40 words) related to your manuscript; suggestions should be sent to Nature Ecology & Evolution as electronic files (the image should be 300 dpi at 210 x 297 mm in either TIFF or JPEG format). Please note that such pictures should be selected more for their aesthetic appeal than for their scientific content, and that colour images work better than black and white or grayscale images. Please do not try to design a cover with the Nature Ecology & Evolution logo etc., and please do not submit composites of images related to your work. I am sure you will understand that we cannot make any promise as to whether any of your suggestions might be selected for the cover of the journal.

You can generate the link yourself when you receive your article DOI by entering it here: <http://authors.springernature.com/share>.

[REDACTED]

P.S. Click on the following link if you would like to recommend Nature Ecology & Evolution to your librarian <http://www.nature.com/subscriptions/recommend.html#forms>

** Visit the Springer Nature Editorial and Publishing website at http://editorial-jobs.springernature.com?utm_source=ejp_NEcoE_email&utm_medium=ejp_NEcoE_email&utm_campaign=ejp_NEcoE for more information about our career opportunities. If you have any questions please click [here](mailto:editorial.publishing.jobs@springernature.com).**